# Design and Experimental Characterization of Artificial Neural Network Controller for a Lower Limb Robotic Exoskeleton

Chih-Jer Lin *[ID] and Ting-Yi Sie

Graduate Institute of Automation Technology, National Taipei University of Technology, Taipei 10608, Taiwan
* Correspondence: cjlin@mail.ntut.edu.tw

**Abstract:** This study aims to develop a lower limb robotic exoskeleton with the use of artificial neural networks for the purpose of rehabilitation. First, the PID control with iterative learning controller is used to test the proposed lower limb robotic exoskeleton robot (LLRER). Although the hip part using the flat brushless DC motors actuation has good tracking results, the knee part using the pneumatic actuated muscle (PAM) actuation cannot perform very well. Second, to compensate this nonlinearity of PAM actuation, the artificial neural network (ANN) feedforward control based on the inverse model trained in advance are used to compensate the nonlinearity of the PAM. Third, a particle swarm optimization (PSO) is used to optimize the PID parameters based on the ANN-feedforward architecture. The developed controller can complete the tracking of one gait cycle within 3.6 s for the knee joint. Among the three controllers, the controller of the ANN-feedforward with PID control (PSO tuned) performs the best, even when the LLRER is worn by the user and the tracking performance is still very good. The average Mean Absolute Error (MAE) of the left knee joint is 1.658 degrees and the average MAE of the right knee joint is 1.392 degrees. In the rehabilitation tests, the controller of ANN-feedforward with PID control is found to be suitable and its versatility for different walking gaits is verified during human tests. The establishment of its inverse model does not need to use complex mathematical formulas and parameters for modeling. Moreover, this study introduces the PSO to search for the optimal parameters of the PID. The architecture diagram and the control signal given by the ANN compensation with the PID control can reduce the error very well.

**Keywords:** pneumatic artificial muscles (PAMs); neural network control; artificial neural network; iterative learning controller; lower limb robotic exoskeleton robot

## 1. Introduction

To perform task-oriented rehabilitation treatment for patients, a variety of robot systems for different purposes and of rehabilitation parts have been developed. The goals of robot systems are to perform specific movements to stimulate the patient's movement plasticity. To achieve the recovery of motor function or minimize the functional deficit of patients, many types of lower extremity rehabilitations have been proposed. The lower extremity rehabilitation system can be mainly classified into the following: (1) Treadmill gait trainer, (2) Footboard-based gait trainer, (3) Ground gait trainer, (4) Fixed gait trainer and (5) Ankle rehabilitation system [1]. Traditional therapies usually focus on treadmill training to improve the functional mobility [2]. This rehabilitation technique is known as partial body-weight support treadmill training. The therapists are required to assist the patient in walking on the treadmill with the legs and hips assisted when the patient's body weight is carried by hanging load belts. For example, the robotic orthosis Lokomat is an automated treadmill training system, which consists of a treadmill and a suspension system to provide the body-weight unloading [3]. The Lokomat consists of a robotic gait orthosis and an advanced body weight support system which is combined with a treadmill. It uses computer-controlled motors for each of its hip and knee joints and the drives are precisely synchronized with the speed of the treadmill to ensure that the speed of the gait orthosis

and treadmill match. The LokoHelp (LokoHelp Group) is an electromechanical device developed for improving gait after brain injury and it is placed on a treadmill parallel to the walking direction to drive the patient to walk [4]. ReoAmbulator (Motorika Ltd., U.S.A., marketed in the USA as the "AutoAmbulator"), is another body-weight-supported treadmill robotic system and it is located in the front of the treadmill and has a protruding link to support the lower limb mechanism [5]. The mechanical lower limb is tied to the patient's leg and there is also a safety strap on the top to support the patient's weight.

In recent years, with the development of neural network related research, many applications have emerged. In the field of controllers, many researchers have been developing systems to make the system more intelligent and able to adapt to complex control principles. Among them, the architecture driven by pneumatic artificial muscles (PAMs) has been a major subject of nonlinear control for many years. Among many PAMs, McKibben Muscle is more commonly used and widely known. It is a type of Braided muscle, and is composed of an air-tight elastic in the middle. The tube is the center, and the elastic tube is surrounded by a braided mesh. When the inner tube is pressurized and inflated, it expands and squeezes the braided mesh. This driving method enables PAMs to have the characteristics of small size, light weight and high output, which is very suitable for the field of rehabilitation robot driving. PAMs have been applied to the development of powered lower limb exoskeletons. For example, Beyl et al. presented a performance evaluation result of a powered knee exoskeleton [6]. The control of PAM-driven systems has proven difficult due to the nonlinear nature of the actuator and the properties of the air pressure source driving it. The model-based control strategies rely heavily on the accuracy of the model to eliminate nonlinearities. Traditional methods such as modeling hysteresis have considered as control pressure, the hysteresis phenomenon and the braided sheath initial angle. However, PAM and many PAM-driven systems generate complex nonlinear forces when pressurized [7,8]; they usually require a lot of time and effort to model the system (which usually requires empirical methods). In addition, the established system model is less resistant to environmental changes or external disturbances. Carbonell et al. [9] discussed the benefits of using three controllers in the pneumatic muscle actuator, namely robust backstepping, adaptive backstepping and sliding-mode. In the study, the tracking is well achieved by the sliding-mode and the adaptive controller. Unfortunately, properties such as PAM actuator dynamics, pneumatic/mechanical system dynamics, and payload characteristics are unknown and/or time-varying.

In many cases sliding mode control may suffer from the same problems as pure model-based control. Feedback error learning (FEL) was originally proposed by Kawato [10]. It is a method to update the feedforward controller through the output error of the feedback controller to improve the accuracy of the inverse model. There has been related discussion about FEL and nonlinear adaptive controllers [11]. However, few FEL concepts are used in the application of PAMs-actuated bidirectional (antagonistic) actuation architecture. To overcome the above-mentioned problems in PAMs modeling, Robinson et al. [12] compared three control strategies: sliding mode control, adaptive sliding mode control, and adaptive neural network (ANN) control. The results show that the ANN controller is preferable because it does not require a model of the pneumatic system or joint mechanism design, which can be difficult and time consuming to characterize, and is robust to changes in PAM actuator characteristics. In this study, a treadmill-type rehabilitation equipment was developed. The rehabilitation movements are used for two kinds of feedforward controllers, including Iterative Learning Control (ILC) and ANN feedforward controllers.

Modeling of PAM-driven rehabilitation machines has been a difficult problem in the field of rehabilitation. On the problem side, the three main challenges proposed in this study are as follows.

1. The complexity of modeling the dual PAM drive (antagonistic) actuation architecture used in this study is relatively high

2. The PAM driver used in this study is a proportional valve, which is cheaper than the pressure control valve, but will increase the complexity of the system.

3. The walking speed set in this study is relatively fast, and it is crucial to overcome the hysteresis problem of PAM, which is also a difficulty point of traditional modeling.

The data collection method used in this study directly oscillates the system through open loop control to obtain the relationship between the knee joint angle and the control command of proportional valve. In other words, we overcome the problems of 1 and 2 by using the forward-feeding ANN, and we verify the operational reliability of the system by conducting experiments on the real system.

On the technical side, there are two novelties.

1. We implemented experiments directly on our LLRER. The PSO-PID controller with a simple feedforward ANN can also obtain good tracking results by sending out the setpoint 3 sampling points ahead of the loop-oriented task.

2. We compensate the PSO-PID controller by using a queue, so that the feedforward ANN does not need the same update frequency as PSO-PID, providing a new option for future integration of other algorithms that cannot be applied to the controller due to the slow update frequency.

## 2. Rehabilitation System Architecture

### 2.1. Design of the Lower Limb Robotic Exoskeleton

Many research laboratories and companies are working on robotic exoskeletons with the intent to assist disabled individuals [10–16]. According to the structural form, lower extremity robotic exoskeletons can be classified into two types: Rigid Lower Extremity Robot Exoskeletons (RLEEX) and Compliant Lower Extremity Robotic Exoskeletons (CLEEX) [17]. Through the RLEEX research, the Human Universal Load Carrier (HULC) of Lockheed Martin [18] and the Guardian XO of Sarcos Robotics [19] in the United States have been the leaders in the development of exoskeletons. Lockheed Martin launched the HULC based on the BLEEX results and conducted a series of wearable tests with the US Army [20]. The Hybrid Assistive Limb (HAL) of the University of Tsukuba adopts a function-oriented design concept; the HAL series [21–24] for medical rehabilitation has been used in Japan and Europe and is the earliest commercial walking exoskeleton robot [25–28].

One of the most-established exoskeleton technologies for disabled assistance is the Rewalk [29]. Robotic exoskeletons can be categorized into three categories according to their purpose. The first group is human efficiency enhancement exoskeletons. The second group involves assistive devices for people with movement disorders due to stroke, spinal cord injury and muscle weakness. The third category is called therapeutic exoskeletons which are utilized for rehabilitation purposes. The first group aims to maximize the durability, stamina, and other physical abilities of persons and is also called augmentation exoskeletons. They may be employed for assisting with lifting heavy items or transporting heavy loads over long distances in manufacturing facilities, urgent relief functions, or military bases. According to the body part involved, the robotic exoskeletons can be categorized into three different categories: upper limb, lower limb and specific joint exoskeletons [14–16]. One of the most significant hurdles to be alleviated is the human-robot interaction and control. Different techniques have been presented in the literature to manage the human-robot interaction.

In this paper, the proposed lower limb robotic exoskeleton is designed for knee and hip joints. One joint has one degree of freedom and the limit of the thigh is designed in the range from $-40$ to $130$ degrees, so that patients can wear the exoskeleton to perform squatting movements. As shown in Figure 1, a DC brushless motor is fixed on the upper side to drive the hip joint; the two PAMs are equipped on both sides to drive the knee joint as shown in Figures 2 and 3. In terms of mechanism design, we installed the PAMs on both sides of the thigh to drive the knee joint. There is a connecting piece between the hip and the back frame, it can adjust the position of the hip joint according to the user's waist circumference (up/down, left/right, front/rear). As shown in Figure 2, the hip flange face is directly connected to the thigh connecting plate and the DC motor (Maxon EC60flat) with the harmonic drive (CSG-17-100-2UH-LW) is used to drive the hip joint. Then, the

thigh connecting plate is connected downward to the leg link, which is used to adjust the length of the thigh.

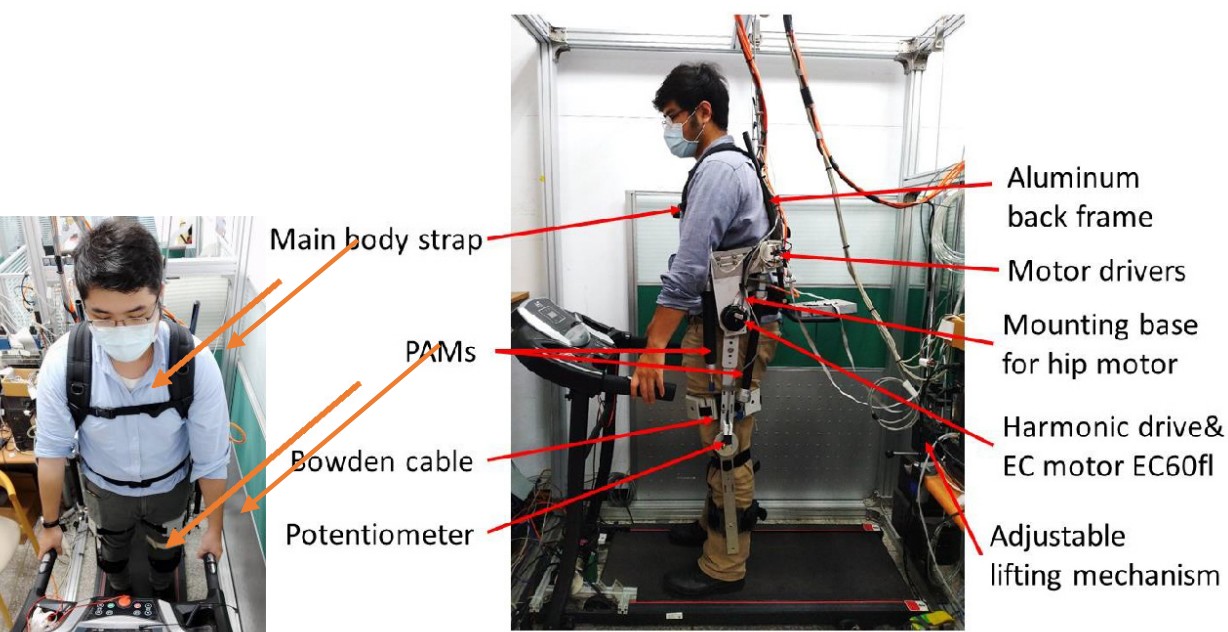

**Figure 1.** Overview of the lower-limb rehabilitation system.

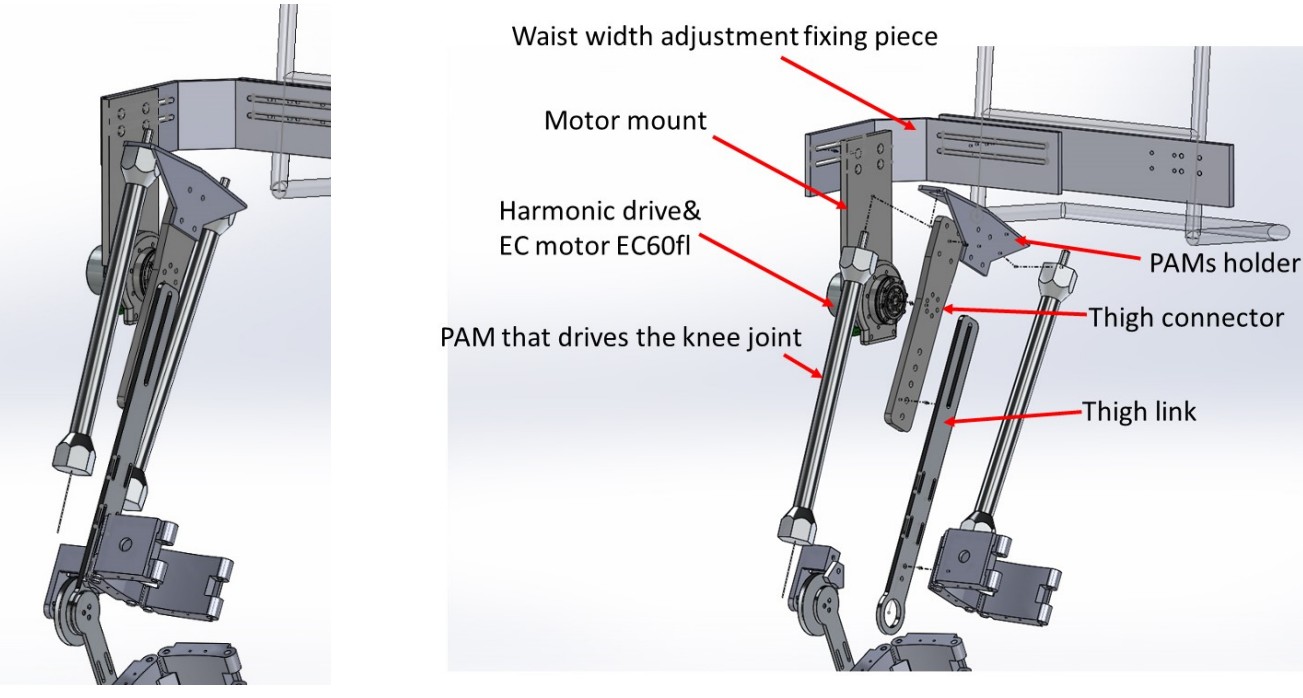

(**a**) Hip exoskeleton.  (**b**) Explosion diagram of the hip exoskeleton.

**Figure 2.** Hip joint exoskeleton design.

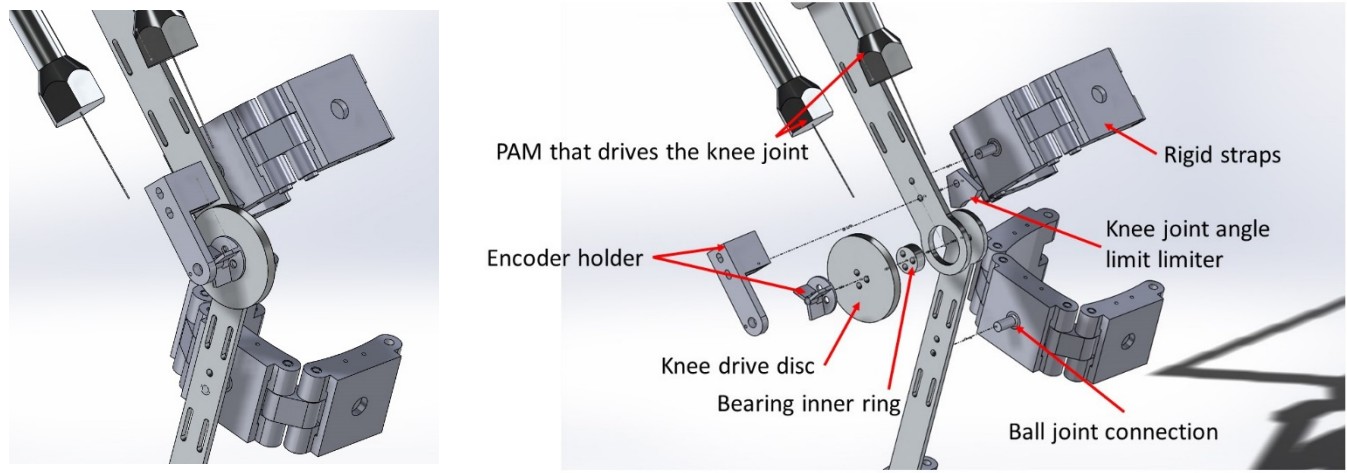

(**a**) Joint exoskeleton.　　　　　　　　　　(**b**) Explosion diagram of the knee exoskeleton.

**Figure 3.** Knee joint exoskeleton design.

As shown in Figure 3, from the design of the knee joint mechanism, the movement of the knee joint comes from the drive disc which is pulled by the two PAMs. The wire is used to maintain the tension pulled by the two PAMs and the proportional directional valve is used to control the contraction and release of the PAMs. In the knee mechanism, a limiting mechanism is used to limit the rotation angle of the knee joint and the rotation range is designed from −10 to 90 degrees. The fixing strap is fixed on the leg, as shown in Figure 3, and the connection part with the mechanism uses a ball joint, so that the rigid strap fits the shape of the subject to a certain extent, and has better rigidity than a pure cloth strap. As shown in Figure 4, the thigh length adjustment mechanism can be adjusted from the shortest distance of 37.2 cm to the longest distance of 52 cm, which can meet the thigh length range of most people. The joint part uses a potentiometer to measure the joint angle.

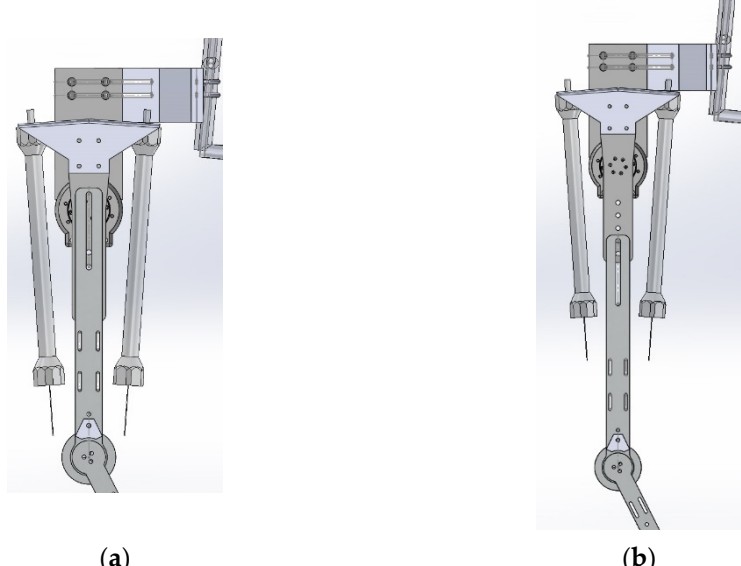

(**a**)　　　　　　　　　　　　　　　　　　(**b**)

**Figure 4.** Thigh length adjustment: (**a**) 37.2 cm; (**b**) 52 cm.

*2.2. Electromechanical System of Powered Lower Limb Rehabilitation Exoskeleton Robot*

This research develops a powered exoskeleton system which has two degree-of-freedom lower limb power exoskeletons, as shown in Figure 5. The hip joint uses a

brushless DC motor with a reducer (Maxon EC60flat + CSG-17-100-2UH-LW) for positioning control as shown in Figure 5. The knee joint uses two PAMs to drive, with driving architecture as shown in Figure 6. The whole system of the proposed lower limb rehabilitation exoskeleton robot system (LLRER) is shown in Figure 7 and all hardware and equipment are used for the LLRER listed in Table 1. The proposed LLRER system is controlled by CompactRIO SbRIO-9631 (National Instrument) with NI 9516 modules, which are responsible for receiving the encoder signal of the motor (Maxon EC 60 flat) with feedback for the current angles of the hip joint to sbRIO-9631 for calculation. In this study, the knee joint is driven by two PAMs (Festo, Germany, MAS-20, as shown in Figure 5) and a proportional directional control valve (Festo, MPYE-5-M5-010-B) is used to control the two PAMs.

　　The knee joint is controlled by the bidirectional actuation via the two PAMs to exert force in two directions, respectively. The proportional directional control valve is operated by converting the voltage input signal to flow directional control signal. The valve is used to control the opening area as well as the inlet and outlet direction through the input voltage to achieve the purpose of controlling the valve. Compared with the single PAM system in which the restoring force comes from gravity or spring, this control method can generally obtain greater joint torsional rigidity, thereby achieving more accurate tracking control results. After the controller algorithm is calculated, the control signal is used to control the knee joint and the hip joint through analog output to achieve the control of the system. The airflow direction of the pressure source is controlled by the proportional directional control valve. The air pressure value and the joint angle value are feedbacked to the embedded controller. The position control PID outputs a directional valve control voltage of 0 V to 10 V, which controls the stretching and contraction of the PAM, and returns the position through the potentiometer of the knee joint as shown in Figure 5.

　　In terms of research and development, PAM is well known to have a relatively small volume ratio while having a high output force. In terms of safety, the shrinkage limit of PAM is about 25%, which is relatively safe, although it is difficult to model, but it has a certain degree of stretching and elasticity, so it is still popular in the field of rehabilitation, providing a certain degree of comfort for the rehabilitation. For the discussion of the controller, we also used PAM drive at the hip joint in the previous research. In the case of fast walking (1 km/h), the PAM response of the hip is not fast enough, so we developed a compound type to support the faster rehabilitation action.

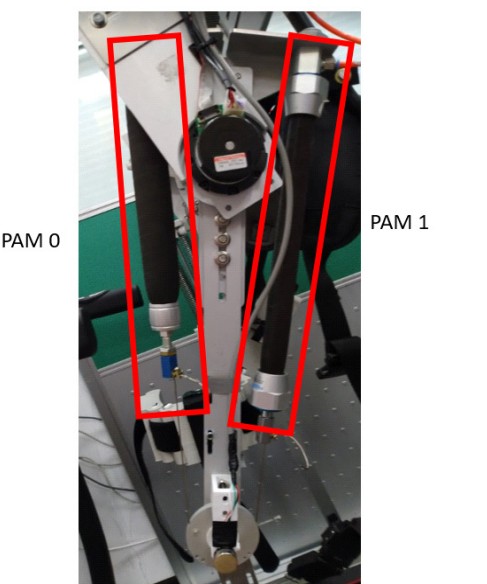
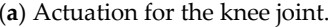
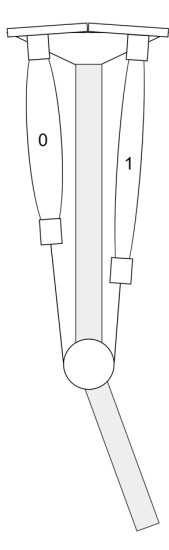

(**a**) Actuation for the knee joint.　　　　　(**b**) Mechanism of the knee joint.

**Figure 5.** Knee joint mechanism.

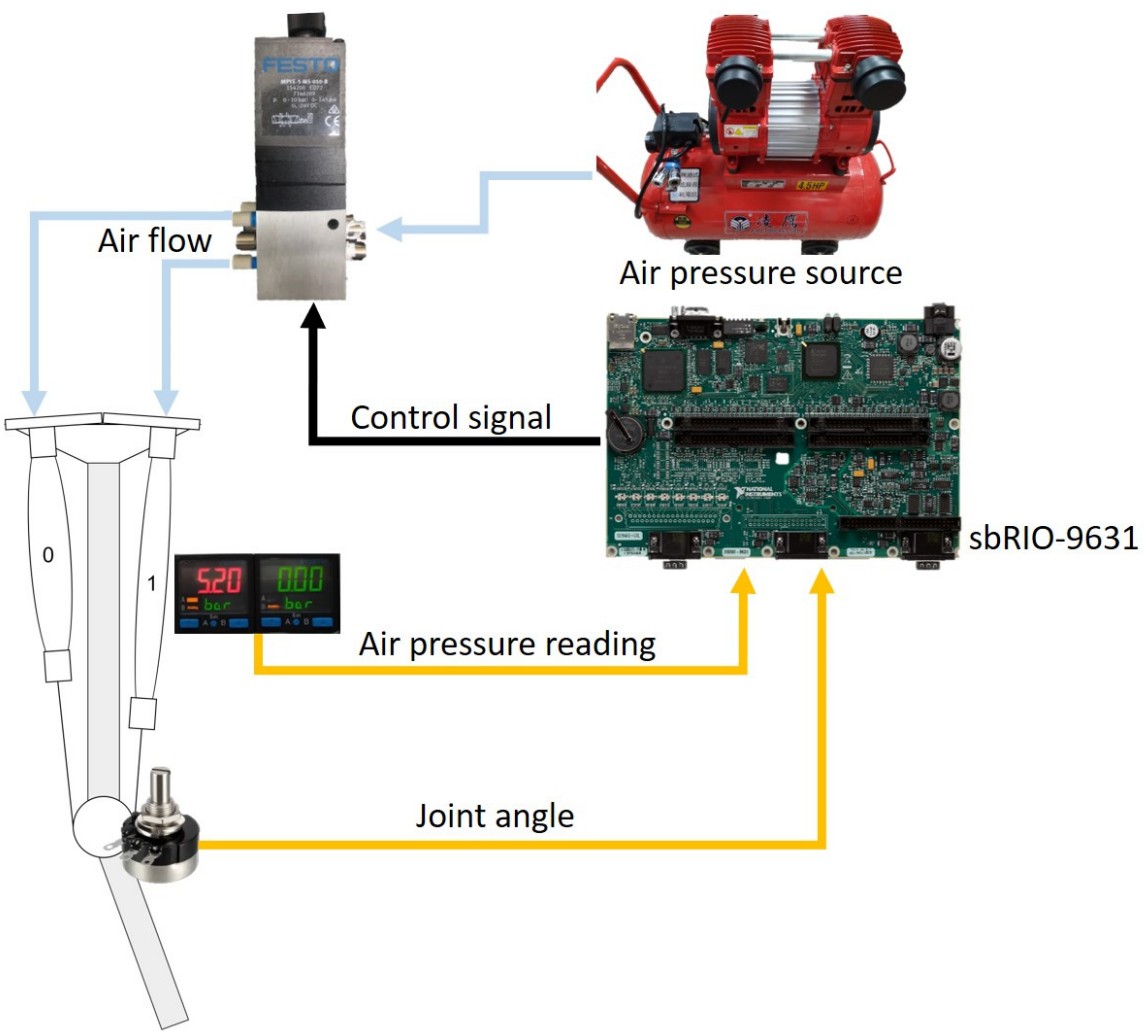

**Figure 6.** Knee joint control architecture.

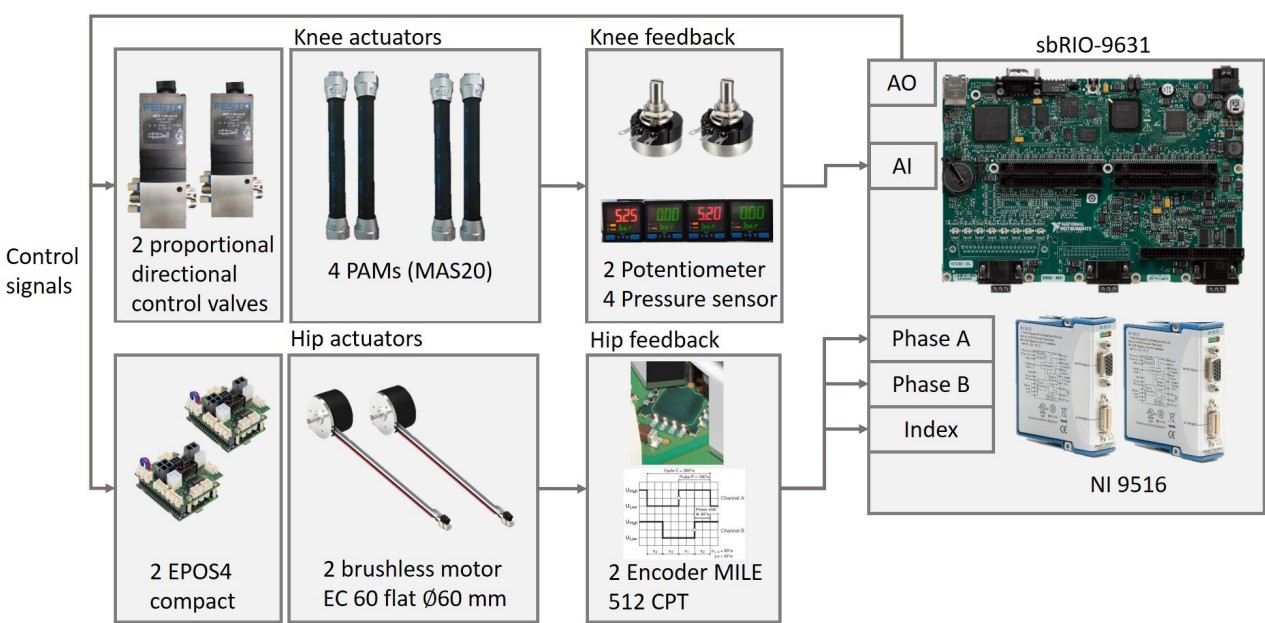

**Figure 7.** Block diagram of the powered lower limb exoskeleton robot.

**Table 1.** Specification list of hardware and equipment for the proposed LLRES.

| Item | Type | Specification |
|---|---|---|
| NI SBRIO-9631 | Embedded controller | Analog&Digital I/O, 266 MHz CPU, 64 MB DRAM, 128 MB Storage, 1 M Gate FPGA |
| NI 9516 | Servo Drive Interface Module | Servo, 1-Axis, Dual Encoder |
| MPYE-5-M5-010-b | Proportional directional control valve | Pressure range: 0~10 bar Input voltage range: 0~10 V |
| MAS-20-300N-AA-MC-O-ER-BG | Pneumatic Artificial Muscle | Operating pressure: 0~6 bar; Maximal permissible contraction: 25% of nominal length |
| Maxon EC60flat | Flat brushless DC motor | Nominal speed: 3730 rpm Nominal torque: 269 mNm |
| CSG-17-100-2UH-LW | Harmonic Drive; with cross roller bearing | Limit for average torque: 51 Nm Limit for Momentary torque:143 Nm |
| SPAB-P10R-G18-NB-K1 | Air pressure sensor | Pressure range: 0~10 bar; Electrical output: 1~5 V analog voltage output |

## 3. LLRER Controller Design

### 3.1. Gait Model Acquisition

To capture the tracking reference of the LLRER system, an unpowered exoskeleton system is made to capture a normal walking reference for the tracking command. As shown in Figure 8, the motion capture system is equipped with 6 sensors on the body. There are two potentiometers on the hip position and the knee joint; a 9-axis IMU (MPU9250) is fixed on the thigh hip to correct the distortion of the hip joint data caused by the back and forth shaking as walking. The sensor signals are captured by the microprocessor (Arduino Uno) for the computation as shown in Figure 9. The IMU is used to transmit the yaw angle from the waist to the hip joint to the PC through I2C; the embedded system converts the potential angular positions of the hip and knee joints into the rotation angle directly. The sampling time of this data collector is 16.3 ms, and the average precision is 0.23 degrees.

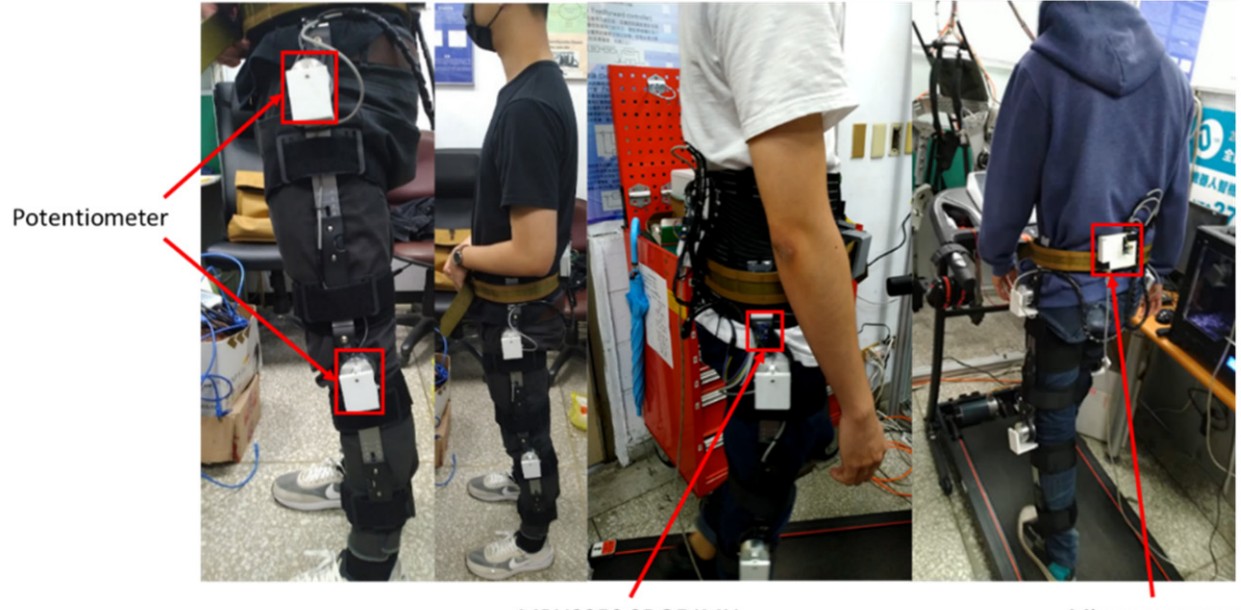

**Figure 8.** Wearing an unpowered exoskeleton.

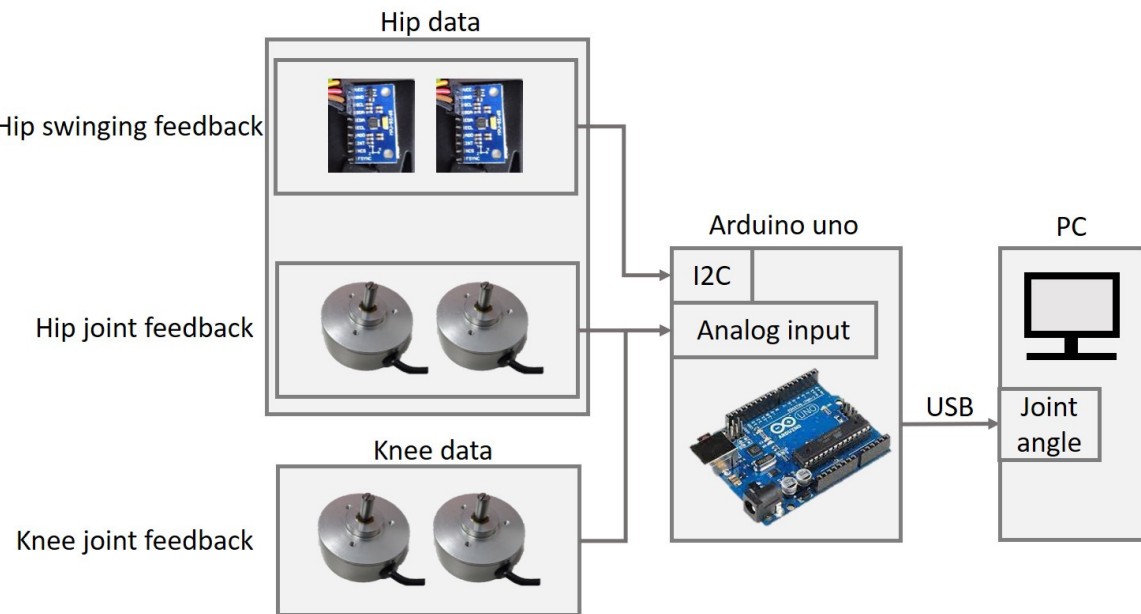

**Figure 9.** Unpowered exoskeleton communication architecture.

The captured angles are filtered and added to the embedded processor; then, the sorted individual gait models are as shown in Figure 10, where V1 represents the walking model at a treadmill speed of 1 km/h and V4 represents a model at the speed of 4 km/h. The data in Figure 10 is the gait motion model at different walking speeds. The gait model is obtained by averaging the trajectories of each walking speed and curve fitting the average trajectory. The gait model is resampled directly to the desired control frequency at the time of use.

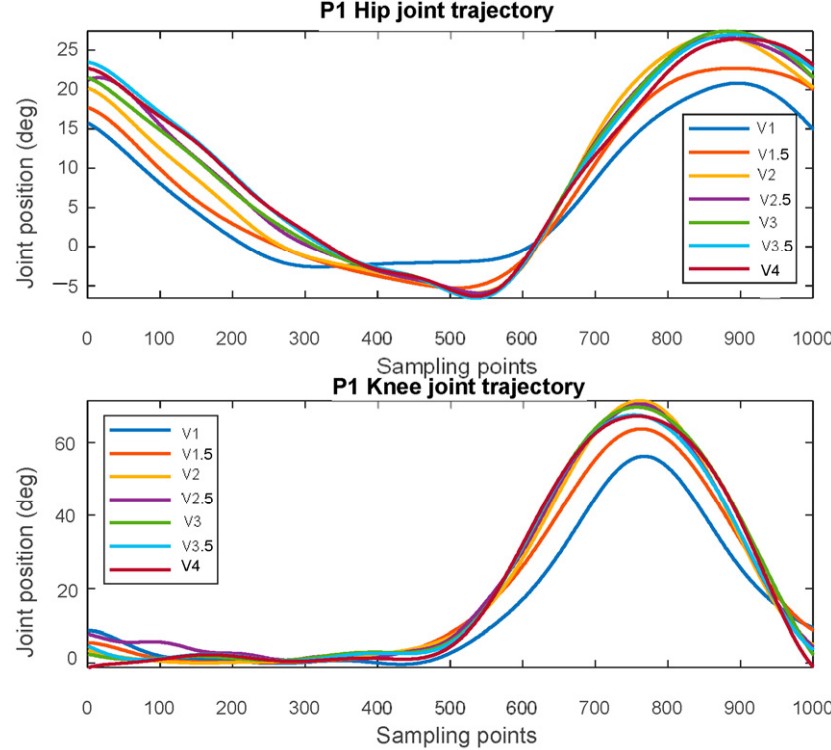

**Figure 10.** Gait model of the P1 subject.

### 3.2. Iterative Learning Control for the LLRER

The Iterative Learning Control (ILC) is an effective control method to improve the tracking error of the repetitive operation of dynamic systems; the rehabilitation gait and movements are usually repetitive movements. The ILC is different from other learning control strategies, such as adaptive control, Repetitive Control (RC) and Neural Networks. The adaptive method modifies the controller itself, while the ILC modifies the output of the controller which reduces the tracking error by changing the control signal. After adjusting the controller using the Ziegler–Nichols method, the tracking error is corrected by the ILC. The definitions of each variable are shown in Table 2. The ILC iteration is calculated in matrix form and the expected trajectory matrix $Y_d$ is determined by the previous measurement. The definition of ILC is shown in Equation (1), where the error of this cycle (the gait cycle) $e_{k \times N}$ is the difference between the expected trajectory matrix $Y_d$ and the real output matrix $y_{k \times N}$. Then the error is multiplied by the learning rate and compensated to the next rehabilitation $\theta_{(k+1) \times N}$.

$$\theta_{(k+1) \times N} = \theta_{k \times N} + L e_{k \times N} \tag{1}$$

$$e_{k \times N} = Y_d - y_{k \times N} \tag{2}$$

**Table 2.** ILC symbol table.

| Notations Type | Specification |
|---|---|
| N | Tracking points per gait cycle |
| $Y_d = (Y_1, \ldots, Y_N)$ | Desired output profile |
| $y_{k \times N} = (y_1, \ldots, y_N)$ | Real output in the current cycle |
| $e_{k \times N} = (e_1, \ldots, e_N)$ | Output error in the current cycle |
| L | Learning rate |
| $\theta_{k \times N} = (\theta_1, \ldots, \theta_N)$ | Control signal of current cycle |
| $\theta_{(k+1) \times N} = (\theta_1, \ldots, \theta_N)$ | Control signal of next cycle |

The control system diagram is shown in Figure 11 and the ILC algorithm updates the desired control signal according to the desired gait. The ILC also compensates the change of the tracking errors, so that the controller can change the control before the change of the tracking error. At first, the ILC is applied to the hip and knee joint control to test the tracking performances. In response, the learning rate L is fixed at 0.02 and the iteration loops are performed 25 times. The same learning rate is used for both the knee and hip joints and the gait model, and then the ILC control experiments are carried out on the knee and hip joints, respectively.

The experimental initial parameters of the PID are obtained through the Z-N method. The controller performance was observed by performing multiple no-load gait experiments at five different speeds, as shown in Table 3. The PID parameters of the hip joint measured by the Z-N method are designed as P: 1.397, I: 0.004, D: 0.001 in the experiments; these PID parameters are used for both hip joints. Figure 12 shows the tracking response of the left and right hip joints using the ILC with the PID learning at a treadmill speed of 1 km/h. Comparing with the results of Figure 12, the ILC can compensate the tracking errors and the lowest tracking errors appear after 10 updates at the speed of 1 km/h. From Table 3, it can be seen that the average error is less than 1 degree and the maximum error is less than 2 degrees. In this hip tracking test, the ILC can compensate the tracking error effectively for the rehabilitation tasks.

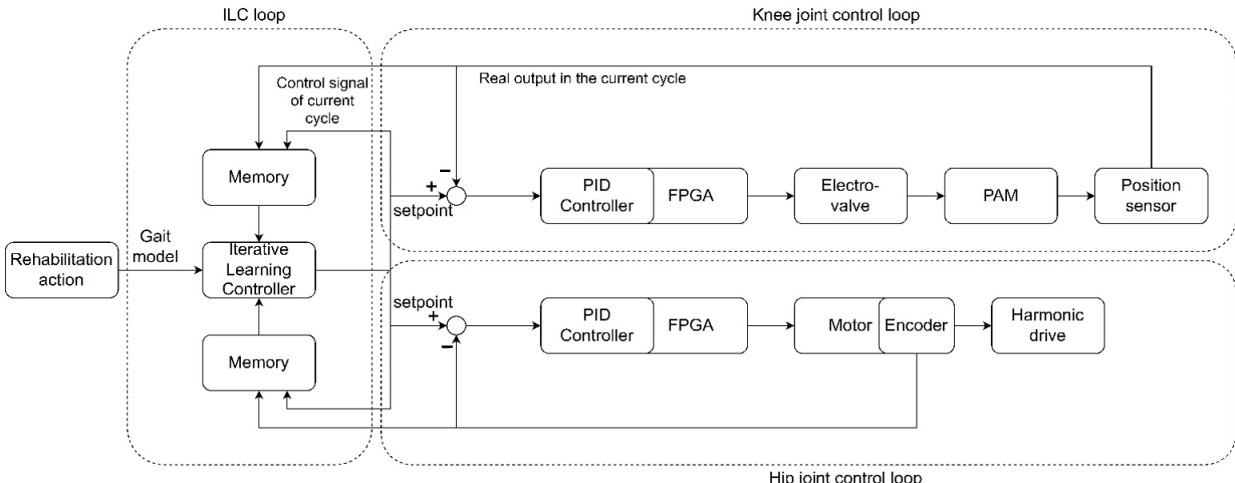

**Figure 11.** Control system of ILC with PID controller.

**Table 3.** Tracking error of hip joint using ILC with PID controller.

| Treadmill Speed (km/h) | Sec/Cycle | Right Hip | | Left Hip | |
|---|---|---|---|---|---|
| | | MAE (°) | MAXE (°) | MAE (°) | MAXE (°) |
| 0.12 | 30 | 0.0241 | 0.5910 | 0.0225 | 0.1280 |
| 0.24 | 15 | 0.0494 | 0.2440 | 0.0440 | 0.2030 |
| 0.53 | 6.8 | 0.1150 | 0.4490 | 0.0890 | 0.4560 |
| 0.85 | 4.25 | 0.3123 | 0.7690 | 0.1856 | 0.7460 |
| 1 | 2.89 | 0.5616 | 1.7750 | 0.4778 | 1.7490 |

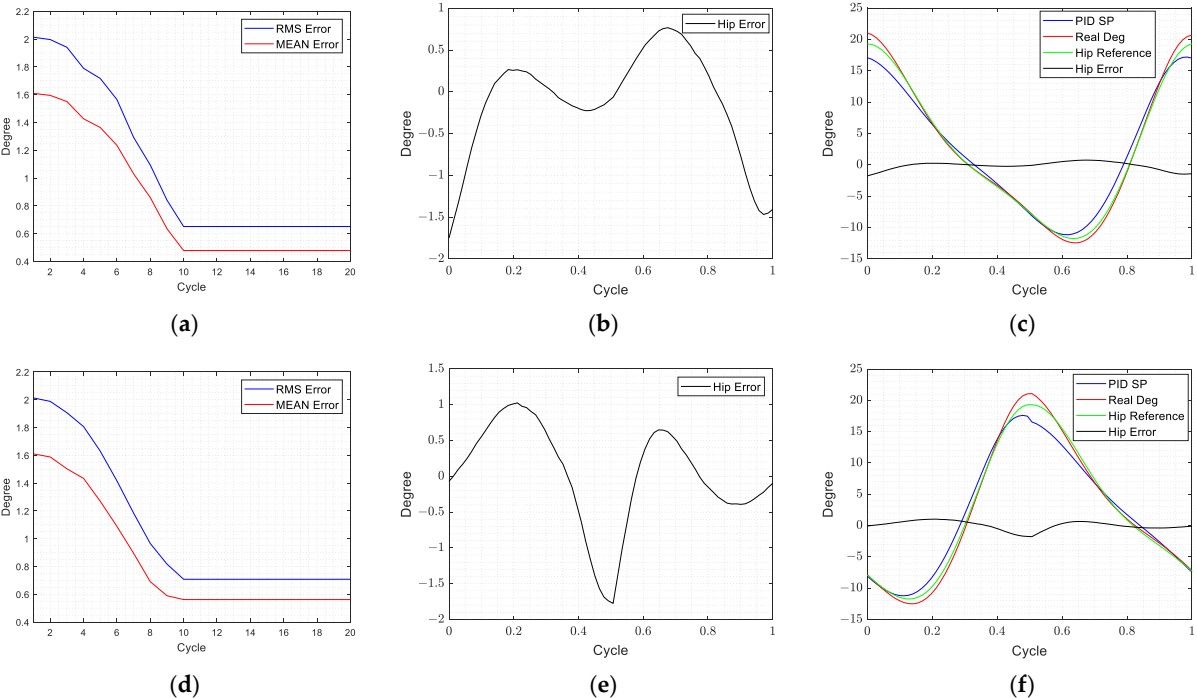

**Figure 12.** ILC with PID learning at a treadmill speed of 1 km/h. (**a**) (Left hip) The number of ILC iterations and the RMS/MEAN error of the trajectory; (**b**) (Left hip) The tracking error diagram of the best gait cycle; (**c**) (Left hip) The tracking response of the best gait cycle.; (**d** (Right hip) The number of ILC iterations and the RMS/MEAN error of the trajectory; (**e**) (Right hip) The tracking error diagram of the best gait cycle; (**f**) (Right hip) The tracking response of the best gait cycle.

After the hip joint test, the knee joint of the proposed system is tested by using the PID controller. The PID parameters (P: 0.203, I: 0.006, D:0.001) are obtained by the Z-N method and the ILC structure is as shown in Figure 13. The knee joints are also tested at five rehabilitation speeds; the tracking results are shown in Table 4. According to the results, the knee joint's response is different from the hip joint, because the use of PAMs gives the system a large tracking error due to the nonlinear characteristics of the PAMs.

The tracking results of the treadmill at 0.85 km/h and 1 km/h are shown in Figure 13 to compare the tracking performance of the PAMs; the SP (setpoint) is the control position command corrected by the ILC controller, Real Deg is the actual response of the system, and knee error is the difference between the knee reference and the actual response of the system. From the experimental results of the system in Figure 13, the knee joint using the PID and ILC cannot achieve performance as the same as the hip joint at the speed of 1 km/h. As the walking speed of the system increases, the effect of the ILC controller is worse. The tracking result of 1 km/h has a large overshoot of the rehabilitation reference trajectory, especially at 0.4 and 0.7 cycles (Figure 13d) and 0.1 and 0.9 cycles (Figure 13b). This indicates that the PAM system needs to find other control methods to compensate it. After using the ILC in the hip and knee joints, it was found that the hip joint could be used with the ILC, while the knee joint needed further improvement. The next section focuses on the improvement of the knee controller.

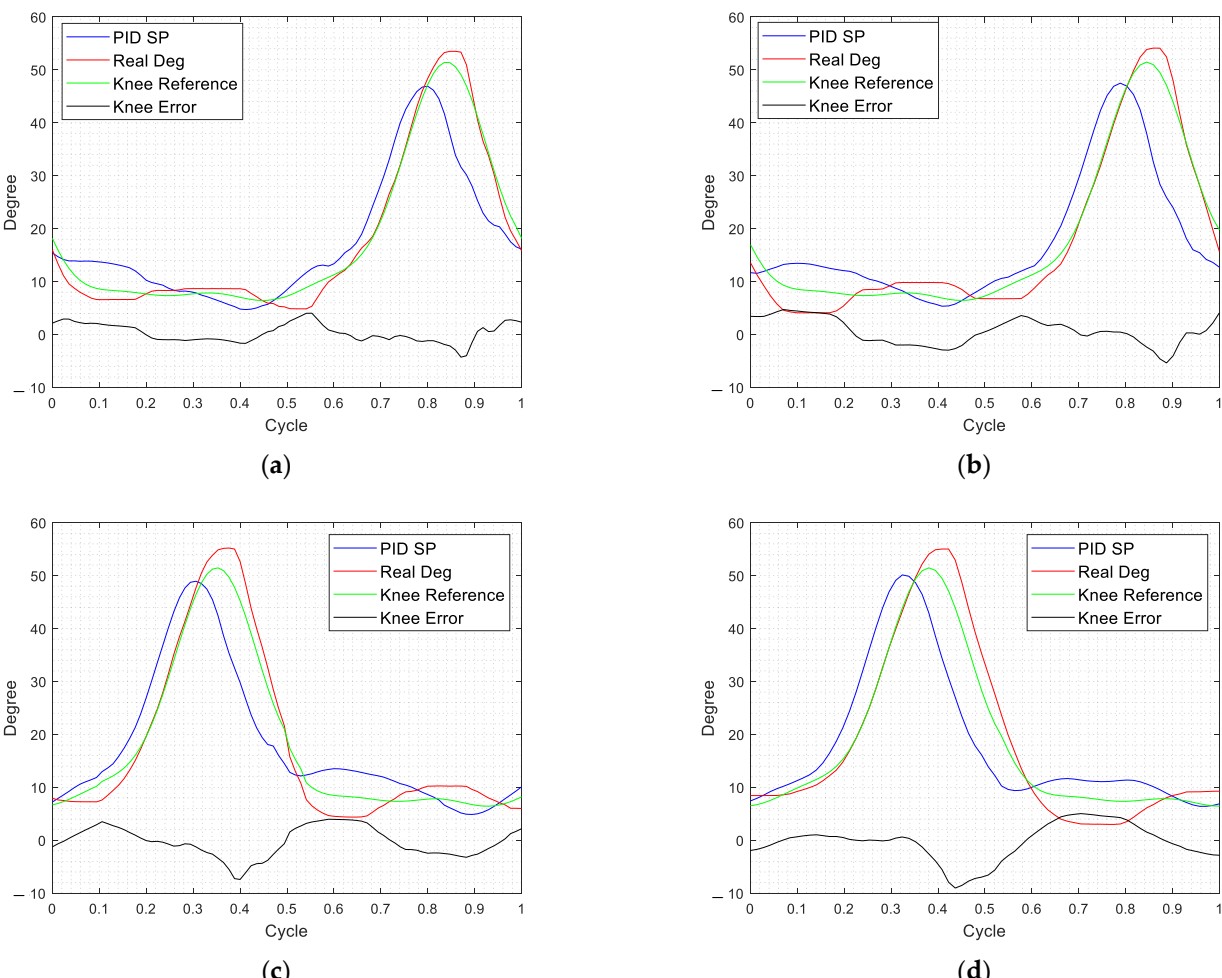

**Figure 13.** Figure **13.** Knee ILC performance at different treadmill speeds. (**a**) (Left knee) Treadmill speed 0.85 km/h; (**b**) (Left knee) Treadmill speed 1 km/h; (**c**) (Right knee) Treadmill speed 0.85 km/h; (**d**) (Right knee) Treadmill speed 1 km/h.

**Table 4.** Tracking error of the knee joint using ILC with PID controller.

| Treadmill Speed (km/h) | Sec/Cycle | Right Knee | | Left Knee | |
|---|---|---|---|---|---|
| | | MAE (°) | MAXE (°) | MAE (°) | MAXE (°) |
| 0.12 | 30 | 0.3944 | 1.9910 | 0.4288 | 1.4910 |
| 0.24 | 15 | 0.9204 | 3.0030 | 0.7004 | 2.4390 |
| 0.53 | 6.8 | 1.1085 | 5.5600 | 0.7162 | 2.7850 |
| 0.85 | 4.25 | 2.3364 | 7.4040 | 1.4856 | 4.2670 |
| 1 | 2.89 | 2.5477 | 9.0250 | 2.1554 | 5.3690 |

## 4. Design of the Feedback Controller for the Knee Joint

### 4.1. Feedforward Artificial Neural Network (ANN) with the Inverse Model

For the network development part, we use Matlab's Neural Net Fitting app for network training, and for the training algorithm, we use Levenberg–Marquardt to update weight and bias values. After training, we integrate the network model into LabVIEW for exoskeleton control. The integration method uses LabVIEW Matlab script to call the established ANN model in the loop of the controller [30–33].

We use the data measured by the real system to train the feedforward ANN controller model in advance. We first set the control command of the proportional directional valve as a linear change in a fixed time, and measure six different time periods to complete a single system response to directional actions. There are two different movements of the knee joint: one is from the straight to the bend (forward movement), and the other is the knee from the bend to the straight (backward movement). Taking Figure 5b as an example, the forward action is PAM0 stretching and PAM1 compression, and the backward action is PAM0 compression and PAM1 stretching. We directly measure a series of system data of these two actions, such as the air pressure of PAM0 (bar) $P_{A0}$ and air pressure of PAM1 (bar) $P_{A1}$ and the knee joint angle $\theta_d$. The time represents that the control signal of the directional valve is sent within 0.5, 1, 2, 3, 4, and 5 s. The corresponding system architecture is shown in Figure 11. Control signals, air pressure readings, and joint angle values are captured while moving, and are used for ANN to learn the system characteristics in advance.

$$\Delta P_{A0} = P_{A0}(n+1) - P_{A0}(n) \tag{3}$$

$$\Delta P_{A1} = P_{A1}(n+1) - P_{A1}(n) \tag{4}$$

$$\Delta \theta_d = \theta(n+1) - \theta(n) \tag{5}$$

$$\Delta cmd = cmd(n+1) - cmd(n) \tag{6}$$

where $P_{A0}(n)$ is the current air pressure (bar) value of PAM0, and $P_{A1}(n)$ corresponds to the air pressure (bar) value of PAM1. $\theta(n)$ is the current knee angle, $cmd(n)$ is the current directional valve control voltage. The data required for training ANN1 (estimating future air pressure changes) can be obtained, and the corresponding current air pressure values $P_{A0}$ and $P_{A1}$, the angle change amount $\Delta\theta_d$ at the next moment, and the corresponding sampling time can be modified according to the delay time that the system needs. The corresponding output is the predicted change in air pressure in the future $\Delta P_{A0}$ and $\Delta P_{A1}$.

To train the ANN1, we use the six experiments to capture the data. Figure 14 shows the six experiments to train the ANN1. Figure 14a,b are the time responses of the $P_{A0}$ and $P_{A1}$ of the knee PAMs with respect to the valve command. Figure 14c represents the knee joint angle with respect to the $P_{A0}$ and $P_{A1}$. The ANN1 is trained with these data to establish the dynamic model. The training set of ANN1 is represented as $TS_{ANN1}$, and the purpose is to give the estimated pressure change with reference to the current system pressure for the input of an ideal angle variation. The collected ANN1 training set is about 3000 sets of input and output corresponding data.

$$TS_{ANN1} = \{P_{A0}, P_{A1}, \Delta\theta_d, \Delta P_{A0}, \Delta P_{A1}\} \tag{7}$$

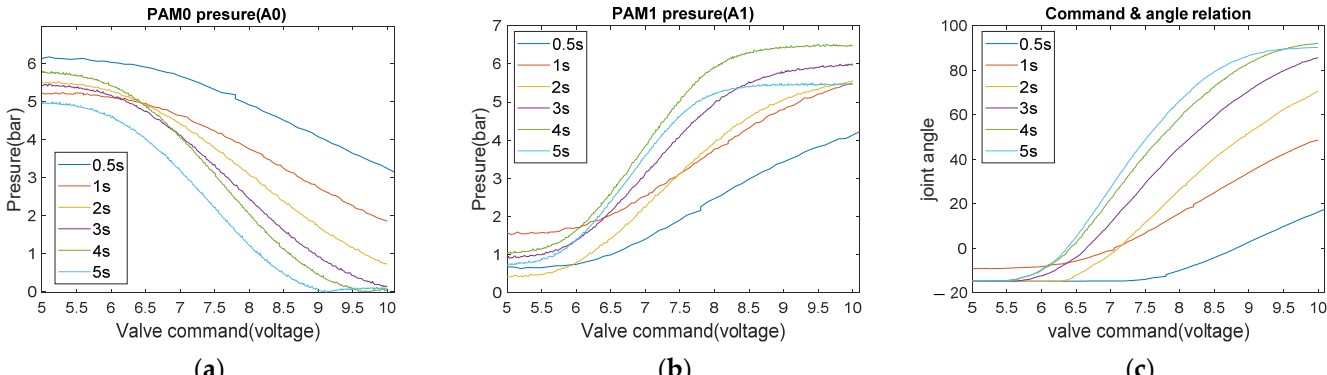

**Figure 14.** Forward movement system response. (**a**) PAM0 pressure changes (bar); (**b**) PAM1 pressure changes (bar); (**c**) command signals and angle relation.

The data for training ANN2 (proportional directional valve) can also be obtained from the same recorded data. The network inputs are the desired angle change $\Delta\theta_d$ and the air pressure change $\Delta P_{A0}$ and $\Delta P_{A1}$ corresponding to the angle change; the output is the corresponding directional valve control voltage change $\Delta$cmd. ANN1 training uses a fully connected network with 3 inputs, 10 hidden layers, and 2 outputs. ANN2 training uses a fully connected network with 3 inputs, 10 hidden layers, and 1 output. This weight is pre-trained and integrated with the controller, as the network is not updated immediately during operation. The collected ANN2 training set has about 1000 input and output corresponding data. The training set of ANN2 is denoted as $TS_{ANN2}$; the purpose is to imitate the model of the proportional directional valve, and convert the air pressure change into control commands.

$$TS_{ANN2} = \{\Delta P_{A0}, \Delta P_{A1}, \Delta\theta_d, \Delta cmd\} \tag{8}$$

Figure 15 uses the feedforward ANN in combination with the PID controller. First, ANN1 refers to the current air pressure A0 and A1 with the desired angle change $\Delta\theta_d$ to predict the expected air pressure change value. ANN2 refers to these air pressure change values and $\Delta\theta_d$ gives a compensated control command $\Delta$cmd, and the tracking trajectory of PID is also advanced by 3 sampling points. The $\Delta\theta_d$ buffer is 10 sampling points in advance, which is a limitation of program development. The prediction time of two pre-trained ANNs integrated into the controller is measured to be 200 ms. In order to make immediate compensation for control commands, it is necessary to predict 4 sets of data at a time to catch up with the time when the ANN runs the next time. It takes 200 ms to wait for 4 data input, and 200 ms to predict, so it is necessary to prepare the ANN data 8 sampling points in advance. Adding the system response delay, the final choice is 10 sampling points in advance.

In other words, the update frequency of the ANN block is 5 Hz, the PID block is 20 Hz, and the control frequency of the exoskeleton is the same as the PID at 20 Hz. ANN predicts 4 pieces of compensation data at a time and queues them at the v buffer. Because the nature of the rehabilitation action is a cyclic action, ANN's queue compensation is feasible. If the controller tracks an acyclic action, the compensation effect of this advance queue may not be ideal. The ideal situation is that there is no need for queues. Here, queues are used because of performance problems in system integration, so the asynchronous method is used.

Figure 16c is the control signal of the feedforward ANN with the inverse model using the PID controller. The ANN trained by using the pre-measured system data can obtain the same control effect. The main control variables are output by the pre-trained network and the PID control is responsible for a small amount of control. The trend of the pressure change predicted by ANN in Figure 16a,b is the same as that of the actual system and the ideal air pressure change is given before the change. The difference of the ideal air pressure

will be compensated with the PID control. From the results shown in Figure 17, the tracking results are good, especially in the area where the tracking angle changes greatly (from 2.5 to 3.6 s). In this experimental result, the performance of the ANN is better than that of the ILC.

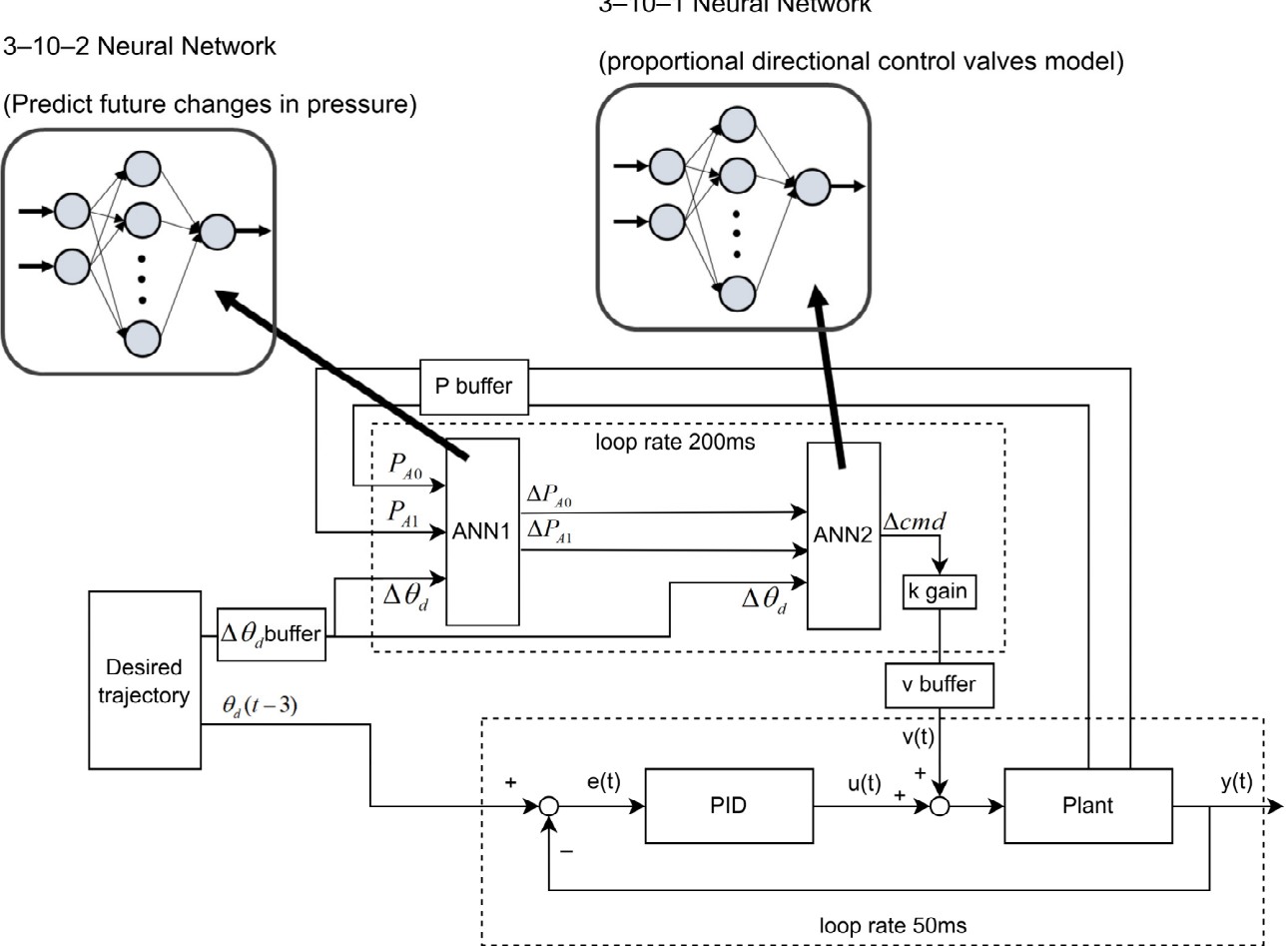

**Figure 15.** ANN-feedforward with PID controller.

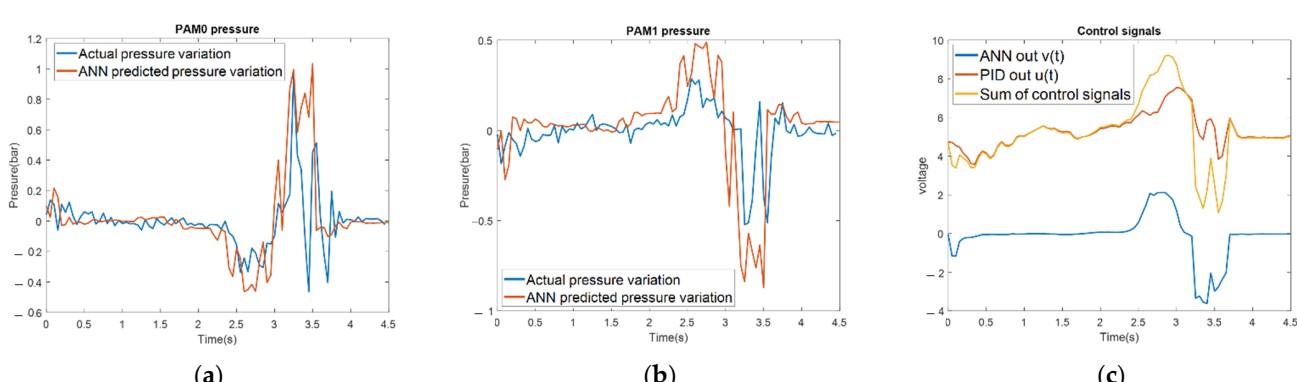

**Figure 16.** Controller signals. (**a**) PAM0 air pressure actual value and ANN predicted value; (**b**) PAM1 air pressure actual value and ANN predicted value; (**c**) control signal of feedforward ANN (IV) with PID.

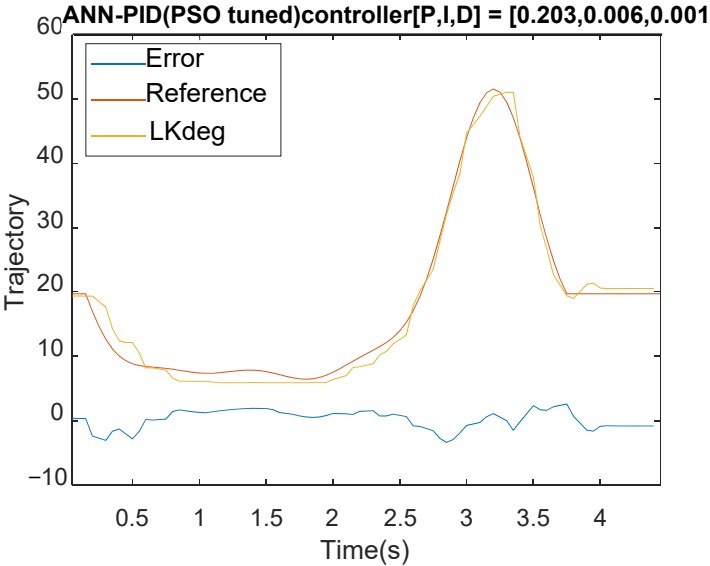

**Figure 17.** Feedforward ANN controller with PID system tracking results.

### 4.2. PSO Tuned PID with ANN Feedforward Control

After the compensation of the ANN feedforward control, the particle swarm optimization (PSO) is used to adjust the parameters of the PID controller. Since the ANN feed-forward has been trained in advance, the next step is to adjust the PID parameters to fit the ANN feedforward controller. Equations (9) and (10) are the calculation methods of the objective function, which are the minimum mean absolute error (MAE) and maximum absolute error (MAXE), respectively, where e is the tracking error of each gait cycle. To minimize MAE and MAXE, the initial individual generation uses the PID parameters obtained by the Z-N method as the initial values (P: 0.203, I: 0.006, D: 0.001); then, the upper limit of the initial population range is set as 0.8~1.2 times the original value. The objective function is set as the sum of 0.7 times MAE and 0.3 times MAXE, as shown in Equation (11). The tracking errors are calculated for each gait cycle and each group of the PID parameters is evaluated for each cycle. The PSO parameter adjustment of the PID parameters is used to test the real system for evaluation. In the PSO method, the population size (popsize) is set to 5 and 20 iterations are performed to find the best parameters. The suitable parameter of interval threshold is set as the 0.2 times of the current best parameter gbest. The update flow chart of PSO is shown in Figure 18 and the overall control flow chart is shown in Figure 19.

$$MAE = \frac{1}{n}\sum_{j=1}^{n}|f_j - y_j| = \frac{1}{n}\sum_{j=1}^{n}e_j \tag{9}$$

$$MAXE = \max_{j=1}^{n}\{|f_j - y_j|\} = \max_{j=1}^{n}\{e_j\} \tag{10}$$

$$\min F = 0.7MAE + 0.3MAXE \tag{11}$$

$$\Delta P_i^{new}(k+1) = w \cdot \Delta P_i(k) + c_1 \cdot r_1 \cdot (pbest_i^p - P_i(k)) + c_2 \cdot r_2 \cdot (gbest^p - P_i(k)) \tag{12}$$

$$\Delta I_i^{new}(k+1) = w \cdot \Delta I_i(k) + c_1 \cdot r_1 \cdot (pbest_i^I - I_i(k)) + c_2 \cdot r_2 \cdot (gbest^I - I_i(k)) \tag{13}$$

$$\Delta D_i^{new}(k+1) = w \cdot \Delta D_i(k) + c_1 \cdot r_1 \cdot (pbest_i^D - D_i(k)) + c_2 \cdot r_2 \cdot (gbest^D - D_i(k)) \tag{14}$$

$$P_i^{new}(k+1) = P_i(k) + \Delta P_i^{new}(k+1) \tag{15}$$

$$I_i^{new}(k+1) = I_i(k) + \Delta I_i^{new}(k+1) \tag{16}$$

$$D_i^{new}(k+1) = D_i(k) + \Delta D_i^{new}(k+1) \tag{17}$$

where Equations (12)–(17) are PSO update formulas for PID parameters; $P_i(k), I_i(k), D_i(k)$ is the position of the i-th particle and the individual in the k-th iteration, $\Delta P_i(k), \Delta I_i(k), \Delta D_i(k)$ is the velocity of the i-th particle and the individual in the k-th iteration; w is the inertia weight; $r_1$ and $r_2$ are two random numbers in the range of 0 to 1; $c_1$ and $c_2$ represent the confidence weight of the particle to itself and the group, generally set from 0 to 4; $pbest_i$ denotes the best position experienced so far by the i-th particle; $gbest$ denotes the best position experienced so far by the entire population.

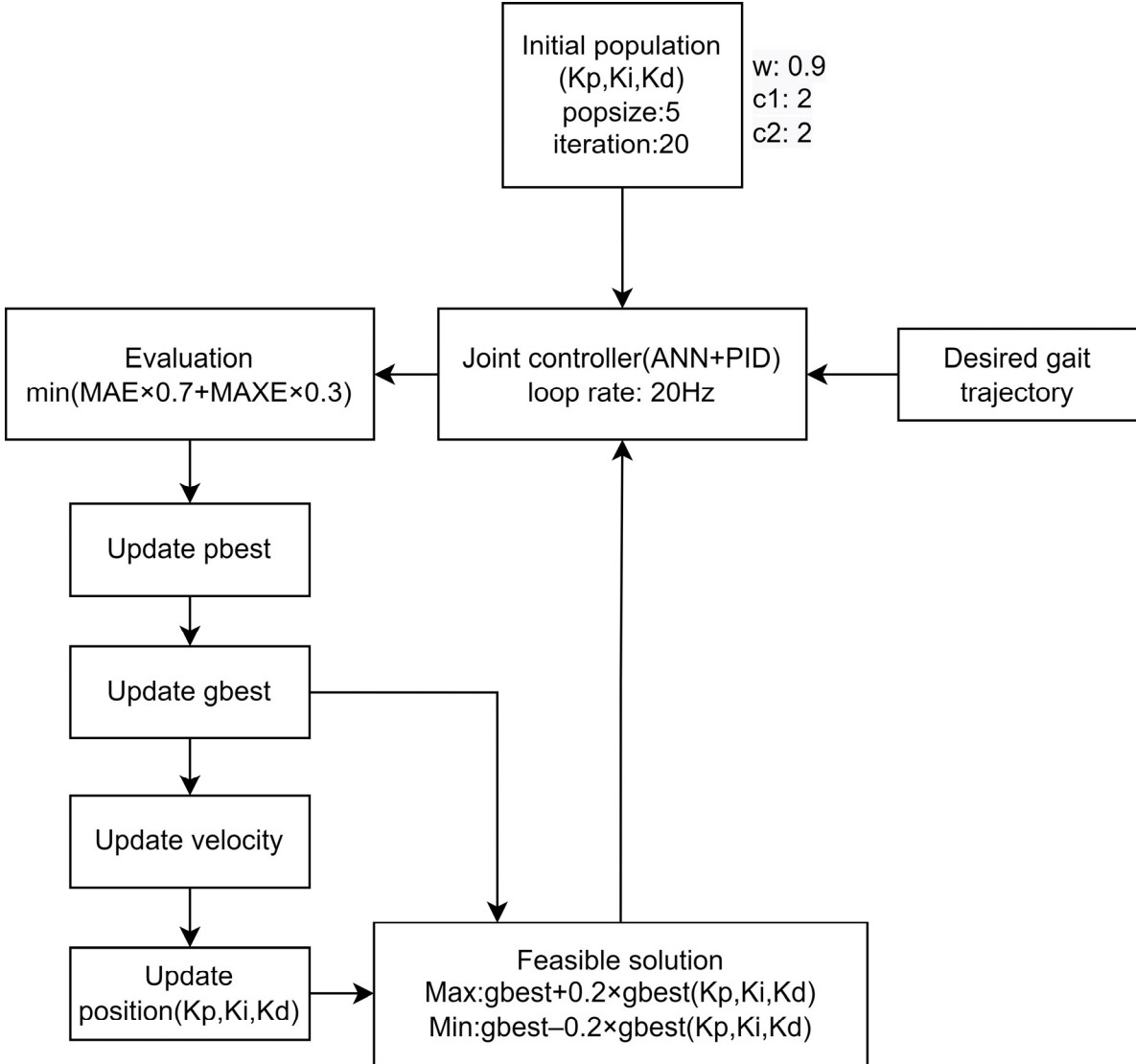

**Figure 18.** PSO tuned PID with ANN flow chart.

When the PSO controller iterates for 20 times, the optimal objective function changes as shown in Figure 20. Figure 21 shows the response with the PID optimization adjustment, after the PSO optimization is performed. With the comparison to Figure 17 (at 1 s), the controller after the PSO adjustment has a better performance than the original in Section 4.1. After the PSO adjustment of the parameters, the MAXE has changed from 4.4 to 3.9 with some overshoot at 3.7 s. Figure 22a shows the difference of the time response for the controllers of PID, ANN + PID, and ANN + PID (PSO adjustment). After adjusting PID parameters, the tracking error is better than the original and the overall MAE decreases as shown in Figure 22b, especially around 1 s. The control signal given by the ANN compensation with the PID can reduce the error very well.

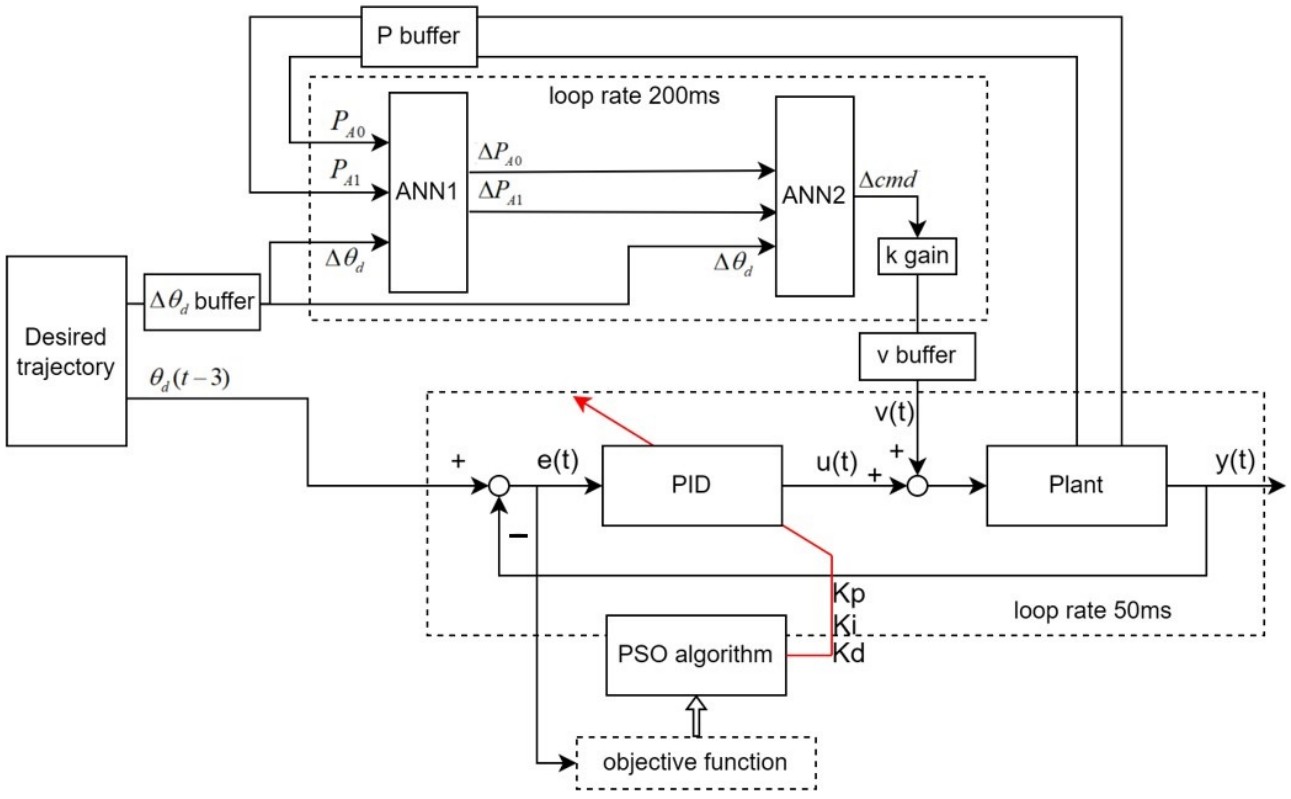

**Figure 19.** PSO tuned PID with ANN controller.

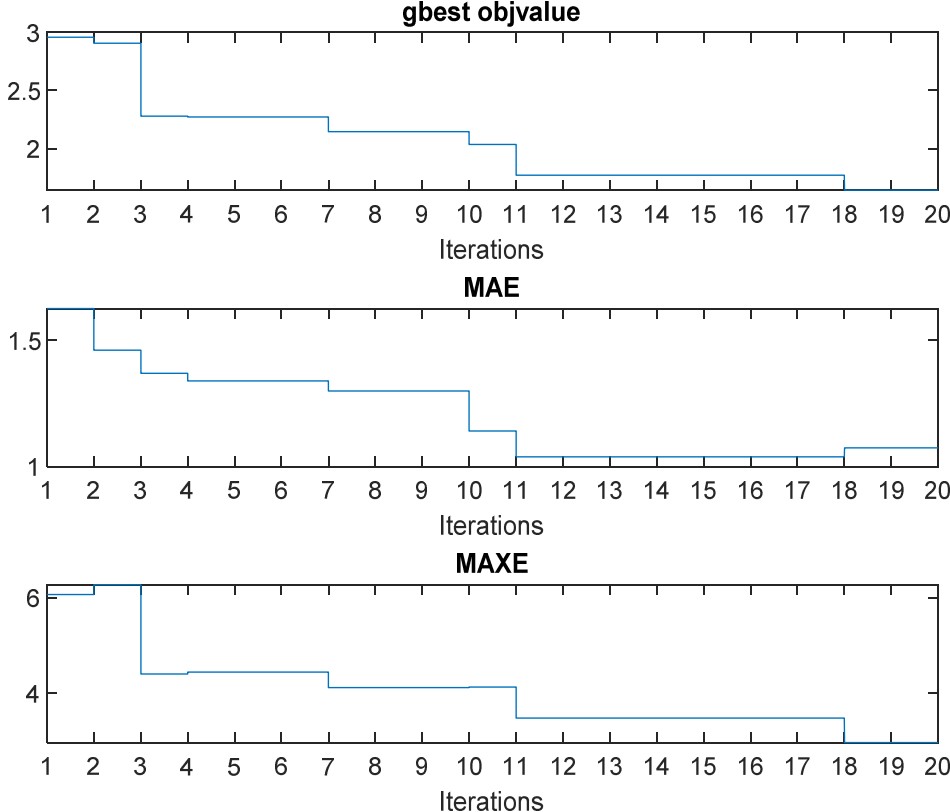

**Figure 20.** PSO iterative parameters.

**ANN-PID(PSO tuned)controller[P,I,D] = [0.295107,0.015306,0.000964]**

**Figure 21.** PSO tuned PID with ANN feedforward control results.

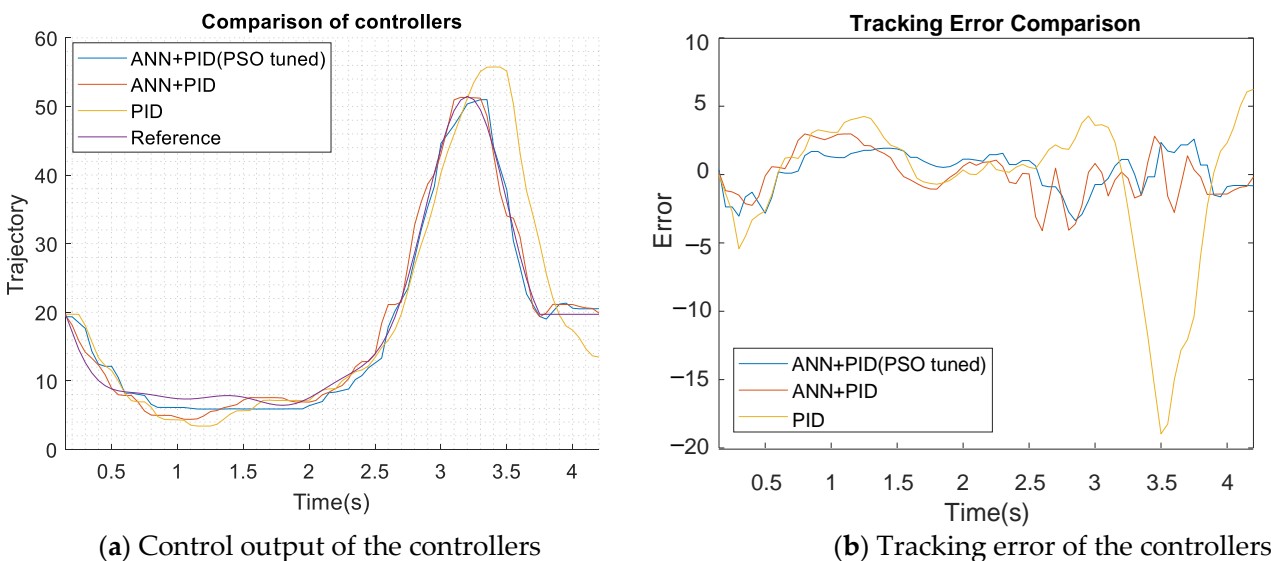

(**a**) Control output of the controllers

(**b**) Tracking error of the controllers

**Figure 22.** PSO tuned controller performance comparison.

## 5. Experiment and Discussion

Previously we discussed three improvements to the knee controller. First, the ILC control architecture is used with the PID control error as the feed-forward update error, expecting to get a good control effect. Secondly, the air pressure and the angle data of different control quantities are collected and the measured data are used to train the inverse

model. The trained control structure (shown in Figure 15) has compensated the tracking error as shown in Figure 17. The third method is to use the PSO to search the optimal parameters of the PID; the architecture diagram is shown in Figure 19. It can be seen that the control signal given by the ANN compensation with the PID control can reduce the error very well.

### 5.1. Knee Joint Controller Performance Comparison

Comparing the effects of the different knee joint controllers, Tables 5 and 6 are the comparative data of the left and right knee joints under the rehabilitation speed of 1 (km/h). According to the experimental results, the PID controller has the worst control response; the feedforward ANN with the PID controller has a better performance than the PID controller; the feedforward ANN with PID (PSO tuned) controller is the best among the three controllers. To test the performance with the subjects, the walking rehabilitation (1 km/h) is performed by the subject. The 1 km/h walking speed is converted into a knee joint cycle time of 3.6 s per cycle, which is relatively fast in the PAM control.

In this experiment, the controller structure is the same as in Figure 15. After the controllers of all joints are integrated into the same program, the operation time of the ANN block is increased from the previous 200 ms to 350 ms due to the computer performance. The buffer size is adjusted from the previous 4 to 7 (350 ms/50 ms) to keep up with the speed of the control loop (50 ms). The controller adjusts the parameters suitable for the current ANN model through PSO and then fixes the optimal parameters. The parameters of the left knee are (P:0.295107, I:0.015306, D:0.000964) and the ones of the right knee are (P: 0.465371, I: 0.017837, D: 0.000236). The control frequency is 20 Hz (sampling time 50 ms) and Figure 23a,b are the experimental result of left knee and the right knee for the PSO tuned PID with ANN feedforward controller. From the experimental results, the on-load tracking error for the proposed controller is still good. In Tables 5 and 6, the MAXE of PSO tuned PID with ANN feedforward is about 3.2 to 6.6 degrees and the MAE is lower than 2 degrees. It can be seen that this control architecture is robust for the subject interference with the system.

**Table 5.** Comparison table of tracking outcomes of different controller (left knee).

| LK | PID | | ANN (Trained IV) + PID | | ANN (Trained IV) + PID (PSO Tuned) | | ANN(Trained IV) + PID (PSO Tuned) with Load | |
|---|---|---|---|---|---|---|---|---|
| Test NO. | MAE | MAXE | MAE | MAXE | MAE | MAXE | MAE | MAXE |
| 1 | 3.091 | 18.381 | 1.425 | 5.273 | 1.226 | 3.680 | 1.870 | 5.336 |
| 2 | 3.665 | 19.497 | 1.480 | 6.481 | 1.214 | 3.976 | 1.575 | 3.524 |
| 3 | 3.388 | 19.282 | 1.199 | 4.426 | 1.195 | 4.275 | 1.608 | 3.849 |
| 4 | 3.325 | 18.329 | 1.257 | 4.099 | 1.237 | 3.357 | 1.333 | 3.174 |
| 5 | 3.590 | 18.961 | 1.217 | 4.728 | 1.181 | 3.933 | 1.901 | 5.348 |

**Table 6.** Comparison table of tracking outcomes of different controller (right knee).

| RK | PID | | ANN (Trained IV) + PID | | ANN (Trained IV) + PID(PSO Tuned) | | ANN (Trained IV) + PID(PSO Tuned) with Load | |
|---|---|---|---|---|---|---|---|---|
| Test NO. | MAE | MAXE | MAE | MAXE | MAE | MAXE | MAE | MAXE |
| 1 | 3.190 | 16.310 | 1.334 | 4.972 | 1.172 | 5.205 | 1.361 | 6.154 |
| 2 | 3.897 | 16.228 | 1.666 | 5.082 | 1.190 | 4.122 | 1.427 | 6.618 |
| 3 | 4.018 | 16.550 | 1.258 | 4.611 | 1.361 | 3.462 | 1.293 | 3.863 |
| 4 | 3.580 | 16.309 | 1.295 | 5.007 | 1.077 | 3.512 | 1.530 | 5.752 |
| 5 | 3.997 | 16.444 | 1.955 | 5.840 | 1.189 | 3.528 | 1.350 | 5.990 |

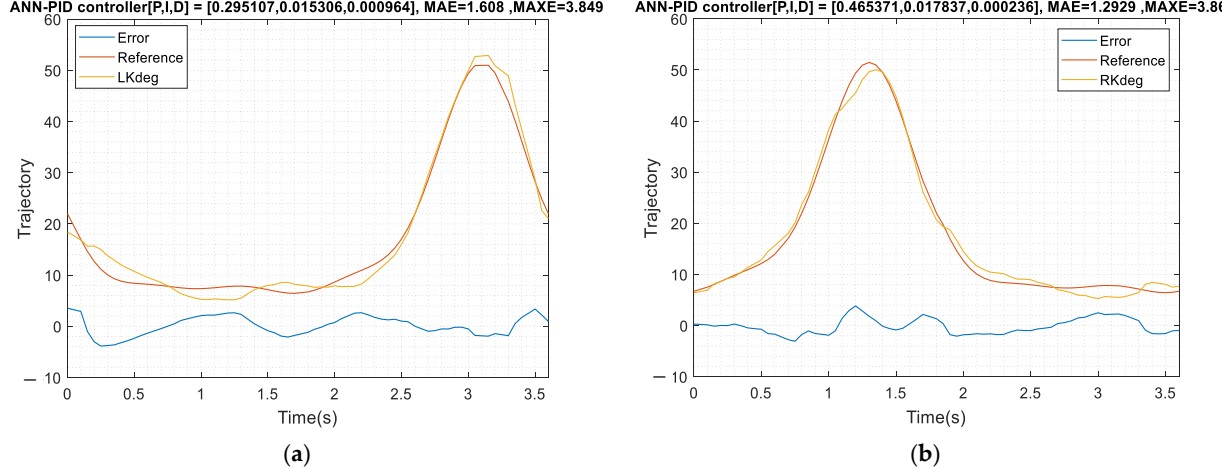

**Figure 23.** PID (PSO tuned)-ANN feedforward control loaded response. (**a**) Tracking result of left knee; (**b**) Tracking result of right knee.

### 5.2. Multi-Subject LLRER Load Experiment

In order to verify the practicability of the proposed PID (PSO tuned)-ANN feedforward controller for the knee joints, an experiment is designed with 10 subjects using the proposed rehabilitation system. In these experiments, the hip joint uses the ILC controller proposed in Section 3.2 and the knee joint uses the PID (PSO tuned)-ANN feedforward controller proposed in Section 5.1. The unique gait models are obtained with an unpowered exoskeleton system as shown in Figure 10 and then the users wear the proposed LLRER for testing. The experimental treadmill speed is set as 1 km/h and the time for one gait cycle is 3.6 s. Both MAE and MAXE are calculated in each gait cycle and the experimental data of the subjects P1 and P2 are shown in Figures 24 and 25.

From the system response of Figures 24 and 25, if the tracking model is replaced with an individual's unique gait, the control strategy proposed can still maintain a good control response. Table 7 shows ten subjects' experimental results and the experimental results show that the controller performs well in the real experiments. The ILC results for the hips show the MAE is 0.915 degrees. In the knee joint experiments using the feedforward ANN with PID (PSO tuned) controller, the average MAE is about 1.66 degrees and the experimental results are also excellent. To indicate the generality of the feedforward controller architecture, the system response data for pre-training is sufficient. The experimental results show that the concept of ANN prediction for this LLRER system is feasible.

**Table 7.** Rehabilitation controller performance data for 10 subjects.

| Loaded Test | Treadmill Speed (1 km/h) | | | | | | | |
|---|---|---|---|---|---|---|---|---|
| | Left_Hip | | Right_Hip | | Left_Knee | | Right_Knee | |
| Controller | PID + ILC | | PID + ILC | | PID (PSO Tuned) +ANN | | PID (PSO Tuned) +ANN | |
| Test NO. | MAE | MAXE | MAE | MAXE | MAE | MAXE | MAE | MAXE |
| P1 | 0.782 | 2.135 | 0.797 | 2.097 | 1.989 | 6.939 | 1.951 | 4.665 |
| P2 | 0.698 | 1.904 | 0.666 | 1.834 | 1.045 | 4.106 | 1.763 | 4.373 |
| P3 | 1.145 | 3.741 | 1.125 | 3.235 | 1.427 | 4.067 | 2.580 | 6.405 |
| P4 | 1.317 | 3.429 | 1.307 | 3.058 | 1.586 | 3.867 | 1.773 | 6.671 |
| P5 | 0.351 | 1.390 | 0.350 | 1.407 | 1.970 | 6.615 | 1.106 | 5.544 |
| P6 | 0.967 | 2.996 | 0.976 | 2.320 | 1.302 | 2.812 | 0.981 | 3.284 |
| P7 | 0.987 | 3.316 | 0.813 | 3.006 | 2.058 | 5.191 | 1.367 | 4.046 |
| P8 | 0.778 | 2.361 | 0.715 | 2.250 | 1.798 | 4.409 | 1.465 | 4.188 |
| P9 | 1.299 | 2.777 | 1.315 | 2.800 | 1.615 | 7.935 | 1.299 | 6.226 |
| P10 | 0.825 | 1.953 | 0.827 | 1.949 | 1.829 | 6.460 | 1.950 | 5.665 |
| avg | 0.915 | 2.600 | 0.889 | 2.396 | 1.662 | 5.240 | 1.623 | 5.107 |



ANN-PID controller[P,I,D] = [0.295107,0.015306,0.000964], MAE=1.9887 ,MAXE=6.939

ANN-PID controller[P,I,D] = [0.465371,0.017837,0.000236], MAE=1.9508 ,MAXE=4.665

(**a**)

(**b**)

PID(ILC) controller[P,I,D] = [1.397,0.004,0.001], MAE=0.78172 ,MAXE=2.135

PID(ILC) controller[P,I,D] = [1.397,0.004,0.001], MAE=0.79683 ,MAXE=2.097

(**c**)

(**d**)

**Figure 24.** Subject P1 data. (**a**) Left knee tracking results; (**b**) right knee tracking results; (**c**) left hip tracking results; (**d**) right hip tracking results.

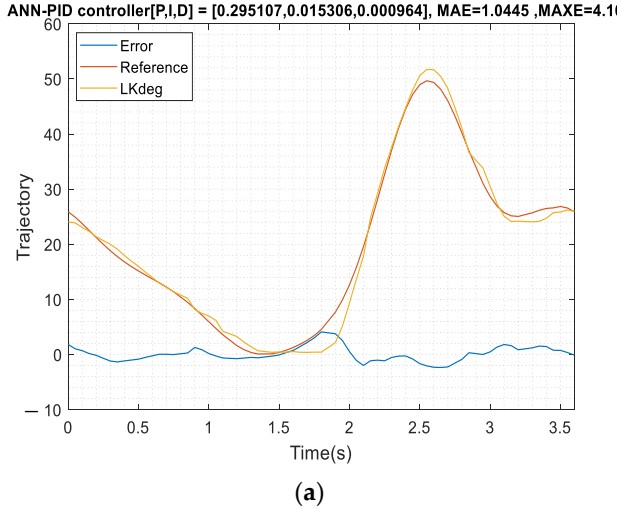

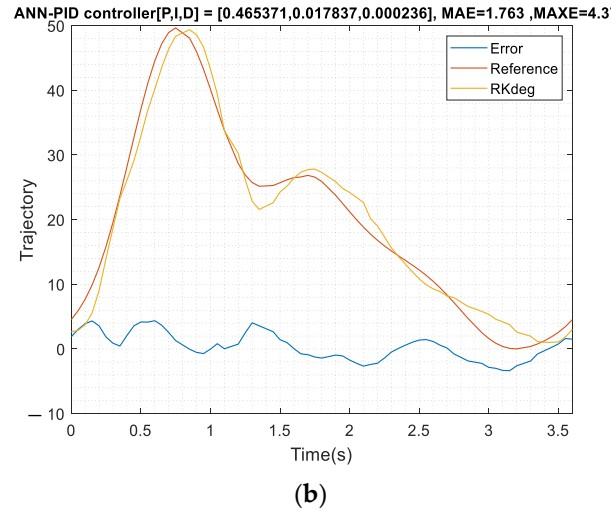

ANN-PID controller[P,I,D] = [0.295107,0.015306,0.000964], MAE=1.0445 ,MAXE=4.106

ANN-PID controller[P,I,D] = [0.465371,0.017837,0.000236], MAE=1.763 ,MAXE=4.373

(**a**)

(**b**)

**Figure 25.** *Cont*.

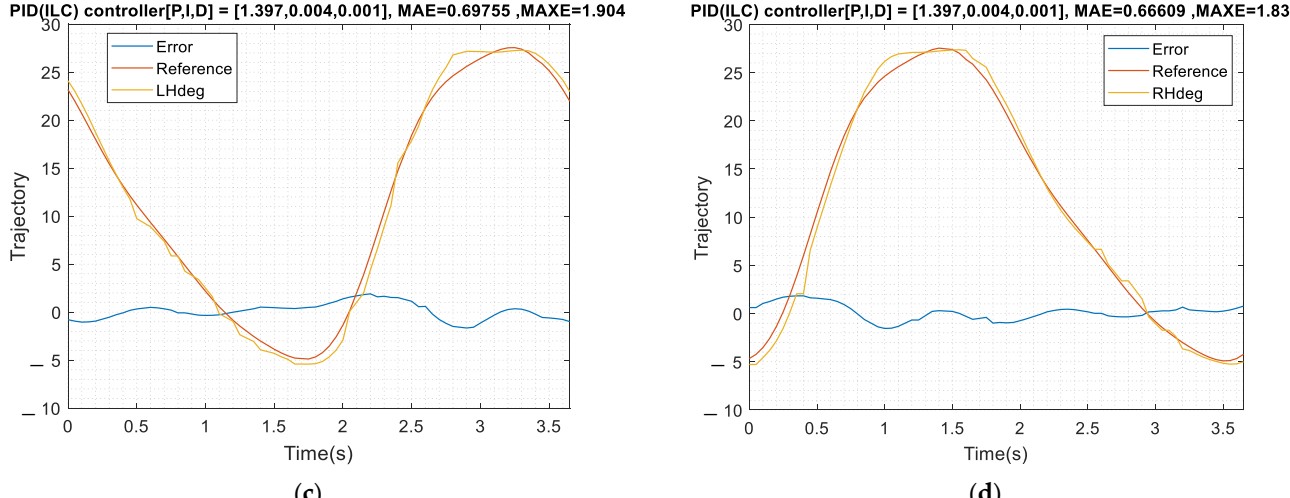

**Figure 25.** Subject P2 data. (**a**) Left knee tracking results; (**b**) right knee tracking results; (**c**) left hip tracking results; (**d**) right hip tracking results.

## 6. Conclusions

In this study, the data collection method of feed-forward ANN is simple. When the system is adjusted due to individual differences of rehabilitation patients, the knee joint only needs to swing back and forth at different speeds to complete the data collection of the new system parameters. There is potential for rapid adaptation in applications. Secondly, the queue method is used to compensate the PSO-PID controller, so that the ANN does not need to update at the same frequency as the PSO-PID, providing a new option for future controller integration. In addition, in the field of lower limb rehabilitation, there are few experimental conditions like ours. Our rehabilitation speed is relatively fast in the application of PAM. The feed-forward ANN combined with PSO-PID can make the performance of the controller on the basis of the traditional Z-N method, and it is optimized to effectively solve the well-known PAM hysteresis problem.

The lower extremity rehabilitation system provides good rehabilitation quality. A DC motor with a reducer for the hip joint and a PAMs-driven bidirectional (antagonistic) actuation for the knee joint are used for the rehabilitation task. First, the ILC algorithm based on the PID controller is used with the feedforward concept and the actual measurement shows that the DC motor of the hip mechanism works well and can provide good rehabilitation (average MAE 0.915 and 0.889 degrees); however, there are nonlinear characteristics for the knee joints and the tracking error is not good enough. Second, to compensate the tracking error of the knee joints, the feedforward concept was used to measure the actual system and the dynamic model was measured by the ANN feedforward control. The air pressure and the angle data of different control quantities are collected and the measured data are used to train the inverse model. The PID controller with the ANN feedforward shows that its response is much better than that of PID. The trained control structure has compensated the tracking error. Third, the PSO is used to search the optimal parameters of the PID and the architecture diagram. It can be seen that the control signal given by the ANN compensation with the PID control can reduce the error very well. The results with the inverse model can be trained with the experimental data without any mathematical modeling. Its versatility for different walking gaits can also be verified during human testing (average MAE 1.66 and 1.623 degrees).

**Author Contributions:** Conceptualization, C.-J.L.; methodology, C.-J.L.; software, T.-Y.S.; validation, C.-J.L. and T.-Y.S.; formal analysis, C.-J.L.; data curation, C.-J.L.; writing—original draft preparation, T.-Y.S.; writing—review and editing, C.-J.L. and T.-Y.S.; visualization, T.-Y.S. and C.-J.L.; supervision, C.-J.L.; project administration, C.-J.L.; funding acquisition, C.-J.L. All authors have read and agreed to the published version of the manuscript.

**Funding:** This research was funded by National Science Council of the Republic of China, grant number Contract No. MOST 108-2221-E-027-112-MY3. The APC was funded by Contract No. MOST 108-2221-E-027-112-MY3.

**Informed Consent Statement:** Informed consent was obtained from all subjects involved in the study.

**Conflicts of Interest:** The authors declare no conflict of interest.

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
