# Peer review of "Design and Experimental Characterization of Artificial Neural Network Controller for a Lower Limb Robotic Exoskeleton"

_actuators, doi:10.3390/act12020055_

Round 1

Reviewer 1 Report

The manuscript entitled Design and Experimental Characterization of Artificial Neural Network Controller for a Lower Limb Robotic Exoskeleton (actuators-2152627-peer-review-v1) present the design of an exoskeleton for therapeutic usage. In particular, the usage of NN controllers is tested and compared with PID controllers. The work is promising; however, several issues prevent the assessment of its true value. Namely:

Abstract – PSO for PID controller tuning is not a new technique, as stated by the authors

L139, Figure 1 – the quality of the picture is very low, some text is unreadable

L145, the exploded view is missing the explosion lines

L153, “as shown in Figure 3, so that the subjects are comfortable while being driven by the exoskeleton.” This is not visible in figure 3

L161, “This research develops a self-developed powered exoskeleton system” please explain!

L199, the system's precision, resolution, sampling time, and repeatability is missing.

L218, Figure 10. The X-axis should be replaced by time

L218, Are the signals filtered?

L263, Figure 11. The block diagram is unreadable thus, it could not be revised

L270, Figure 12. The plot quality is too low to revise.

L277, why use a PI controller?

L281, if so, why use the PAM at all?

L300. Figure 13, the figure is unreadable,

L293, the following chapter? If the performance is low, why not directly present this new controller? Please explain.

L341, Figure 14, the plots font is too small, please revise.

L368, what is the controller executing frequency? 5Hz?

L379, Figure 15, the quality of the figure is not adequate.

L382, Figure 16, the plots font is too small, please revise.

L421, Figure 18, the quality of the figure is not adequate.

L423, Figure 19, the quality of the figure is not adequate.

L405, equations 19-27 are not cited in the document!

L436, Figure 20, the quality of the figure is not adequate.

L440, Figure 21, the quality of the figure is not adequate.

L433, Figure 20, the quality of the figure is not adequate.

L446, the low quality of the figures, prevented an adequate revision of the discussion. As such, this was not considered.

It is unclear why the authors used two different actuators: an electrical motor and pneumatic muscles. Also, is not explained why the author performed initial testing of the system to determine the individual joint curve instead of performing a numeric simulation in Simulink, Openmodelica, or another similar program. Also, with so many controllers in the literature that surpass the PIDs, why use these? The plotting of the error instead of the curves would also help in the comparison of the methods. Overall, the poor quality of the figures and plots prevents a detailed and complete analysis of the work.

Author Response

L139, Figure 1  the quality of the picture is very low, some text is unreadable

Thank you for your comments. The text in Figures 1, 2, and 3 has been replaced with enlarged fonts, and the original pictures are used to enhance contrast and improve clarity.

L145, the exploded view is missing the explosion lines

Thank you for your comments. We have added the original figures and add explosion lines in the original figures (Fig. 3 and Fig.4).

L153, “as shown in Figure 3, so that the subjects are comfortable while being driven by the exoskeleton.” This is not visible in figure 3

Thank you for your comments. We have added the expression to the Fig. 3 and explain how to wear the exoskeleton by the subjects. (Fig3 ball joint connection & L175)

L161, “This research develops a self-developed powered exoskeleton system” please explain!

Thank you for your comments. We have revised the typo by “This research develops a powered exoskeleton system” and ” This rehabilitation system is developed by our team”. Sorry for our error expression. (L184)

 L199, the system's precision, resolution, sampling time, and repeatability is missing.

Thank you for your comments. The system’s precision is about 0.23 degree and the sampling time is about 16.3ms. (L239)

L218, Figure 10. The X-axis should be replaced by time

Thank you for your comments. This is the captured Gait model. Therefore, the X-axis should be the sampling times. (L245)

L218, Are the signals filtered?

Thank you for your comments. The signals are filtered by the curve fitting to find the gait trajectory. (L245)

L263, Figure 11. The block diagram is unreadable thus, it could not be revised

Thank you for your comments. We have revised the figure with a clear font size.

L270, Figure 12. The plot quality is too low to revise.

Thank you for your comments. We have revised the Figure 12 with a clear font size.  (fig 12 && table3)

L277, why use a PI controller?

Thank you for your comments. For the common sense, we have revised the controller with PID controller.  (L325 & table4 & fig13)

L281, if so, why use the PAM at all?

Thank you for your comments. For the research consideration, the PAM has the high torque with the minimal space. Moreover, it has the flexible properties for users. Therefore, we design the PAM system for the knee actuation. (L210)

L300. Figure 13, the figure is unreadable,

Thank you for your comments. We have revised the Figure 13 with a clear font size.  

L293, the following chapter? If the performance is low, why not directly present this new controller? Please explain.

Thank you for your comments. In this chapter, we find the hip joint can be controlled by the ILC; however, the knee joint can not be controlled by the ILC. Therefore, we try another controller (ANN+PID) to compensate it. (L324)

L341, Figure 14, the plots font is too small, please revise.

Thank you for your comments. We have revised the Figure 14 with a clear font size.  

L368, what is the controller executing frequency? 5Hz?

Thank you for your comments. The controller executing frequency of the ANN is 5Hz; however, the executing frequency of PID is 20Hz. Therefore, the PID controller frequency of the system is also 20Hz. (L407)

L379, Figure 15, the quality of the figure is not adequate.

Thank you for your comments. We have revised the Figure 15 with a clear font size.  

L382, Figure 16, the plots font is too small, please revise.

Thank you for your comments. We have revised the Figure 16 with a clear font size.  

L421, Figure 18, the quality of the figure is not adequate.

Thank you for your comments. We have revised the Figure 18 with a clear font size.  

L423, Figure 19, the quality of the figure is not adequate.

Thank you for your comments. We have revised the Figure 19 with a clear font size.  

L405, equations 19-27 are not cited in the document!

Thank you for your comments. We have cited them in the revised document. (L440 L446)

L436, Figure 20, the quality of the figure is not adequate.

Thank you for your comments. We have revised the Figure 20 with a clear font size.  

L440, Figure 21, the quality of the figure is not adequate.

Thank you for your comments. We have revised the Figure 21 with a clear font size.  

L433, Figure 20, the quality of the figure is not adequate.

Thank you for your comments. We have revised the Figure 20 with a clear font size.  

 (fig22 L483)

L446, the low quality of the figures, prevented an adequate revision of the discussion. As such, this was not considered.

Thank you for your comments. We have revised the figures and made the discussion. (Figs. 23~25 have been revised.)

 It is unclear why the authors used two different actuators: an electrical motor and pneumatic muscles. Also, is not explained why the author performed initial testing of the system to determine the individual joint curve instead of performing a numeric simulation in Simulink, Openmodelica, or another similar program. Also, with so many controllers in the literature that surpass the PIDs, why use these? The plotting of the error instead of the curves would also help in the comparison of the methods. Overall, the poor quality of the figures and plots prevents a detailed and complete analysis of the work.

Thank you for your comments. We have tried our best to revise our paper.

Reviewer 2 Report

The paper submitted for review concerns the use of Artificial NeuralNetwork for Controller for a Lower Limb Robotic Exoskeleton seems to be very interesting and in line with modern trends, however, it seems to me that it is mostly about implementation, and less scientific, when it comes to the scientific aspect, I have a few questions that I think should be explained in the context of the artificial neural networks themselves.

1. Please, write what software or programming language was used to develop artificial neural networks? Write it at the beginning of chapter 4.1.

2. On what basis was it decided to use 10 neurons in the hidden layer? Why such a number and not another?

3. What neural activation functions have been used in neural networks? Some information about this can be found in the articles:

https://doi.org/10.3390/app112110414

https://doi.org/10.3390/app10051897

4. Did you use one of the existing network learning algorithms or did you write your own?

5. In the conclusions, write what is the novelty of your paper compared to other publications.

6. Many pictures are not clear, too low resolution, please correct, for example, picture 20b, picture 13 a, b, c, 12 c, f ....

Author Response

  1. Please, write what software or programming language was used to develop artificial neural networks? Write it at the beginning of chapter 4.1.

Thank you for your comments. In the controller development, we used the Neural Net Fitting App (Matlab) for the neural network. We use Levenberg-Marquardt to update weight and bias values in the program. After the neural networks has been trained, it will be integrated with Labview matlab script to implement the ANN model. (L340)

  1. On what basis was it decided to use 10 neurons in the hidden layer? Why such a number and not another?

Thank you for your comments. The relevant variables that will be affected include the number of network layers, activation function, size of training set, number of neurons in the input layer and number of neurons in the output layer. They will affect the selection of neurons. The number of neurons in the hidden layer is usually 10~40. Due to the computation efficiency, we use 10 neurons to train in the real time implementation.

  1. What neural activation functions have been used in neural networks? Some information about this can be found in the articles:

https://doi.org/10.3390/app112110414

https://doi.org/10.3390/app10051897

Thank you for your comments. We use the Sigmoid function for the activation function. The references [31-32] have been included in the paper.

  1. Did you use one of the existing network learning algorithms or did you write your own?

Thank you for your comments. We use the Levenberg-Marquardt for the network learning algorithm. Based on the experiments, it can be more suitable to our system.

  1. In the conclusions, write what is the novelty of your paper compared to other publications.

Thank you for your comments. We have modified our conclusion to write the novelty of our paper. (L579-589)

  1. Many pictures are not clear, too low resolution, please correct, for example, picture 20b, picture 13 a, b, c, 12 c, f ....

 Thank you for your comments. We have modified the pictures in the revised paper.

Reviewer 3 Report

This paper developed a lower limb robotic exoskeleton with the artificial neural networks for the rehabilitation tasks. However, the current work of this manuscript is not comprehensive enough to be accepted for its publication in the journal, and some major improvements is necessary in the content of the different sections.

1.       In section 1, the necessity and innovations of the paper were not described clearly. ANN, PAM, PSO are widely used in the robotics. Thus, it's not clear what the contribution on both problem side and technical side.

2.       In the introduction, some latest references of upper limb and lower limb rehabilitation robots need to be cited. For example: A review on lower limb rehabilitation exoskeleton robots 2019

Author Response

  1. In section 1, the necessity and innovations of the paper were not described clearly. ANN, PAM, PSO are widely used in the robotics. Thus, it's not clear what the contribution on both problem side and technical side.

In the field of rehabilitation, the modeling of PAM-driven rehabilitation machines has always been a difficult problem. The three main difficulties overcome in this study are as follows. (1) The modeling complexity of the dual PAM drive (antagonistic) actuation architecture used in the study is relatively high. (2) The PAM driver used in this study is a proportional valve, which is more expensive than pressure control Valve is cheap, but increases system complexity. (3) The walking speed set in this study is relatively fast, and whether the lag problem of PAM can be overcome is more critical here, which is also the difficulty of traditional modeling.

In this study, the data collection method used is to directly obtain the relationship between knee angle and proportional valve control; in other words, the proposed control overcomes the problems of (1) and (2) via the ANN forward controller to verify the reliability of the system by experimenting with the real system.

In the technical side, the novelty of this paper can be divided into two points as follows.

(1) By experimenting directly on our LLREL, the loop-oriented task is sent out of the sampling points in advance and the PSO-PID controller can also obtain good tracking results with the ANN feed-forward control.

(2) We use the queue method to compensate the PSO-PID controller, so that the feedforward network does not need to update the same frequency as the PSO-PID. We provides a new option for future integration which the other algorithms cannot be applied to the controller due to slow updates. (L98)

  1. In the introduction, some latest references of upper limb and lower limb rehabilitation robots need to be cited. For example: A review on lower limb rehabilitation exoskeleton robots 2019

Thank you for your comments. We have cited the latest reference [30] in our paper.

Round 2

Reviewer 1 Report

The manuscript has been improved, in particular, the quality of the figures.

An ethical issue that is not addressed relates to the test subjects. Did the test subject adequately inform of the procedure/test and its goal? Do the authors have the informed consent documents?

Author Response

Q: An ethical issue that is not addressed relates to the test subjects. Did the test subject adequately inform of the procedure/test and its goal? Do the authors have the informed consent documents?

A: Thank you for your comments. This project cooperates with Dr. Si-Huei Lee of Taipei Veterans General Hospital to develop rehabilitation technology. Before the experiment, each subject is informed of the complete experimental process, as well as the experimental objectives, and wears rehabilitation equipment. The task content guided by the researcher, such as: walking, fast walking, slow walking, and the action content related to evaluate the rehabilitation ability. Participants have the right to suspend and withdraw from the experiment at any time for any reason, and they are also informed of the risks of the experiment and signed the relevant documents before the experiment.

Reviewer 2 Report

The article has been corrected. If you can please add also to references https://doi.org/10.3390/app122111160 best regards

Author Response

Q: The article has been corrected. If you can please add also to references https://doi.org/10.3390/app122111160 best regards.

A: Thank you for your comments. We have added the reference in [33]. 

Reviewer 3 Report

The revised contents of the paper are perfect.

Author Response

Thank you for your comments.